# TIMING: Temporality-Aware Integrated Gradients for Time Series Explanation

**Hyeongwon Jang** [* 1]  **Changhun Kim** [* 1 2]  **Eunho Yang** [1 2]

## Abstract

Recent explainable artificial intelligence (XAI) methods for time series primarily estimate point-wise attribution magnitudes, while overlooking the directional impact on predictions, leading to suboptimal identification of significant points. Our analysis shows that conventional Integrated Gradients (IG) effectively capture critical points with both positive and negative impacts on predictions. However, current evaluation metrics fail to assess this capability, as they inadvertently *cancel out* opposing feature contributions. To address this limitation, we propose novel evaluation metrics—Cumulative Prediction Difference (CPD) and Cumulative Prediction Preservation (CPP)—to systematically assess whether attribution methods accurately identify significant positive and negative points in time series XAI. Under these metrics, conventional IG outperforms recent counterparts. However, directly applying IG to time series data may lead to suboptimal outcomes, as generated paths ignore temporal relationships and introduce out-of-distribution samples. To overcome these challenges, we introduce TIMING, which enhances IG by incorporating temporal awareness while maintaining its theoretical properties. Extensive experiments on synthetic and real-world time series benchmarks demonstrate that TIMING outperforms existing time series XAI baselines. Our code is available at https://github.com/drumpt/TIMING.

## 1. Introduction

Time series data are prevalent across various fields, especially in safety-critical domains such as healthcare (Sukkar et al., 2012; Bica et al., 2020; Rangapuram et al., 2021), energy (Rangapuram et al., 2018; Benidis et al., 2022),

---
[*]Equal contribution [1]Kim Jaechul Graduate School of AI, KAIST, Daejeon, South Korea [2]AITRICS, Seoul, South Korea. Correspondence to: Eunho Yang <eunhoy@kaist.ac.kr>.

*Proceedings of the 42nd International Conference on Machine Learning*, Vancouver, Canada. PMLR 267, 2025. Copyright 2025 by the author(s).

transportation (Nguyen et al., 2018; Song et al., 2020), and infrastructure (Dick et al., 2019; Torres et al., 2021). These sectors usually necessitate high transparency in predictive models to ensure safe and reliable operations, making the interpretability of model behavior crucial. With deep neural networks becoming the dominant approach for time series analysis, understanding their decision-making processes has become increasingly challenging due to their black-box nature (Zhao et al., 2023; Liu et al., 2024b;a).

To tackle this challenge, several XAI methods for time series data have been proposed (Tonekaboni et al., 2020; Leung et al., 2023; Crabbé & Van Der Schaar, 2021; Enguehard, 2023; Liu et al., 2024b; Queen et al., 2024; Liu et al., 2024a; Ismail et al., 2020). These methods commonly employ *unsigned attribution schemes*, focusing on the *magnitude* of output changes resulting from feature removal, rather than their *direction*—whether they reinforce or oppose the model's prediction. This is practically undesirable as end-users typically want to identify positively contributing features. Furthermore, existing evaluation strategies assess these methods by measuring prediction changes by *simultaneously masking out the most important points based on high attribution scores*; this approach does not adequately capture the effectiveness of methods such as Integrated Gradients (IG) that provide directional attribution.

Regarding this matter, our preliminary analysis in Section 3.1 reveals that while IG effectively captures important points, this capability has been significantly underestimated in prior studies; this underestimation occurs since traditional metrics cancel out the effects of points with opposing impacts. By relying solely on these evaluations, recent literature has inadvertently favored methods that align attribution directions while neglecting the interpretative value of directional information.

Motivated by these limitations, we first propose novel evaluation metrics—Cumulative Prediction Difference (CPD) and Cumulative Prediction Preservation (CPP)—to comprehensively assess both directed and undirected attribution methods. These metrics evaluate points cumulatively rather than simultaneously: CPD for high-attribution points (higher is better) and CPP for low-attribution points (lower is better). By addressing the *cancel out* problem of positive and negative attribution under these metrics, we re-evaluate existing

baselines and find that traditional gradient-based methods, such as IG, perform remarkably well compared to state-of-the-art methods (Enguehard, 2023; Liu et al., 2024a;b). This result demonstrates that these methods with directional attributions are more effective at capturing the points that truly influence the model's behavior.

Building on the efficacy of directional attribution methods, we propose **Tim**e Series **In**tegrated **G**radients (TIMING), a novel approach designed to enhance conventional IG tailored for the time series domain. While traditional IG calculates gradients along a zero baseline at all points simultaneously, it often fails to capture the effects of complex temporal dependencies and introduces out-of-distribution (OOD) samples along the integration path. TIMING overcomes these challenges by integrating temporality-aware stochastic baselines into the attribution calculation process. We further analyze the theoretical guarantees of TIMING, demonstrating that its segment-based masking can be incorporated into the internal IG path, preserving key theoretical properties of IG while enabling more effective calculation. Extensive experiments on synthetic and real-world datasets demonstrate that TIMING outperforms existing time series XAI baselines.

To summarize, our contributions are threefold:

- We propose CPD and CPP, which monitor all internal changes rather than making simultaneous changes, to resolve the *cancel out* problem present in existing time series XAI evaluations.
- We introduce TIMING, which improves IG by adapting temporality-aware stochastic baselines to capture the effects of complex temporal dependencies and mitigate OOD problems.
- Extensive experiments show that TIMING outperforms baselines while maintaining the efficiency and theoretical properties of IG.

## 2. Problem Setup

We define the problem of estimating feature attribution for time series data. Let $F : \mathbb{R}^{T \times D} \rightarrow [0, 1]^C$ be a time series classifier, where $T$ is the number of time steps, $D$ is the feature dimension, and $C$ is the number of classes. Let $x \in \mathbb{R}^{T \times D}$ denote an input time series. The classifier $F$ outputs class probabilities $F(x) = (F_1(x), \ldots, F_C(x))$ where $F_c(x) \geq 0$ and $\sum_{c=1}^{C} F_c(x) = 1$. A feature attribution $\mathcal{A}(F, x) \in \mathbb{R}^{T \times D}$ is a real-valued matrix, where each entry $\mathcal{A}(F, x)_{t,d}$ quantifies the contribution of the feature $x_{t,d}$ to the model's predicted probability for the chosen class $F_{\hat{y}}(x)$, with $\hat{y} = \arg\max_{c \in \{1, \ldots, C\}} F_c(x)$.

For signed attribution methods (Lundberg & Lee, 2017; Ribeiro et al., 2016; Sundararajan et al., 2017; Shrikumar et al., 2017), positive values of $\mathcal{A}(F, x)_{t,d}$ indicate that the

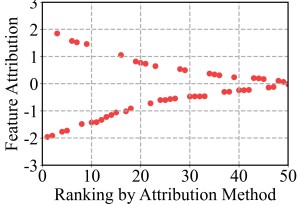 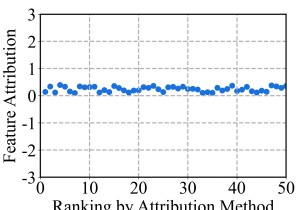

**(a)** Correctly ranked attributions with unaligned signs.  **(b)** Poorly ranked attributions with aligned signs.

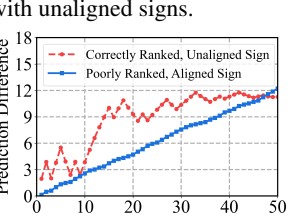 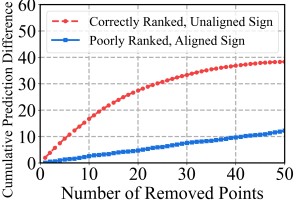

**(c)** Existing raw prediction difference.  **(d)** Proposed cumulative prediction difference.

**Figure 1:** An example illustrating how cumulative prediction difference (CPD) improves upon raw prediction difference. While raw difference incorrectly favors a poorly performing method with aligned signs (blue, b) over a perfect method with misaligned signs (red, a), CPD correctly identifies the superior performance of the latter (d vs. c).

feature $x_{t,d}$ contributes to increasing model's prediction score for the chosen class, while negative values suggested that $x_{t,d}$ suppresses the prediction score, with the absolute value reflecting the strength of influence. In contrast, unsigned attribution methods (Suresh et al., 2017; Tonekaboni et al., 2020; Leung et al., 2023; Crabbé & Van Der Schaar, 2021; Enguehard, 2023; Liu et al., 2024b; Queen et al., 2024; Liu et al., 2024a) focus solely on the magnitude of contributions (*i.e.*, high $\mathcal{A}(F, x)_{t,d}$ indicates the importance of $x_{t,d}$ for the model's prediction), highlighting the relative importance of each feature without indicating the direction of its impact.

## 3. Proposed Metrics

In this section, we first discuss the limitations of the evaluation metrics of current time series XAI algorithms in Section 3.1. We then introduce our novel metrics, Cumulative Prediction Difference (CPD) and Cumulative Prediction Preservation (CPP) in Section 3.2.

### 3.1. Limitations of Current Time Series XAI Metrics

Evaluating XAI methods is challenging due to the absence of a definitive ground truth regarding which parts of the input the model focuses on, especially in modern deep neural networks (Lundberg & Lee, 2017). Yet, these methods should be quantitatively evaluated, and thus we typically consider two approaches: 1) assessing whether high attribution is assigned to the input elements that were used to generate the output in a synthetic dataset, where the data gen-

eration process is known, or 2) measuring the performance drop (*e.g.*, Accuracy, AUROC, or AUPRC drop) when the top-$K$ high attribution points are removed from a real-world dataset. However, in the case of synthetic data, there is no guarantee that the model correctly focuses on the important points dictated by the data generation process. Additionally, performance drop analysis relies on ground truth labels, which shift the focus of the explanation from the model to the data. For instance, when the model has learned spurious correlations or is attending to irrelevant features, removing high-attribution regions can inversely improve performance.

A more reasonable approach in terms of faithfulness is to measure the prediction difference before and after removing the top-$K$ salient points, which can be formulated as:

$$\Delta \hat{y} = \left| F_{\hat{y}}(x) - F_{\hat{y}}(x_K^\uparrow) \right|,$$

where $x_K^\uparrow$ denotes the input time series with the top-$K$ salient points removed, and $|\cdot|$ represents the absolute value. This stems from an intuition that if the model truly depends on certain points, removing them should significantly alter the prediction, while removing irrelevant points should have minimal impact. However, issues arise when multiple top-$K$ points are removed *simultaneously*. As shown in Figure 1, this introduces a *sign-aligning bias*, where features with small but consistently directed attributions are incorrectly deemed important, as positive and negative attributions *cancel out* their influence on the prediction. Specifically, when measuring naive prediction difference (Figure 1 (c)), removing the top 50 attribution points with a poor ranking but aligned signs (Figure 1 (b)) ultimately outperforms the removal of perfectly ranked yet misaligned attributions (Figure 1 (a)). This misdirection can skew time series XAI research toward merely aligning attribution directions.

Our preliminary experiment in Table 1 further demonstrates that aligning positively or negatively attributed points using IG (without taking the absolute value) significantly outperforms existing baselines, further highlighting the flaw in Acc (10%), which is the ratio of retained predictions after masking 10% of points. Consequently, simply applying a ReLU activation to IG's attribution—an approach analogous to GradCAM (Selvaraju et al., 2017) and GradCAM++ (Chattopadhay et al., 2018), two widely adopted XAI methods in computer vision—can misleadingly yield state-of-the-art results under these metrics. However, within the context of simultaneously removing important points, relying solely on these directionally aligned activation maps constitutes an inherently unfair comparison. Based on these findings, we claim that such metrics inherently suffer from a shared limitation, and thereby propose novel evaluation metrics in Section 3.2.

**Table 1:** Preliminary evaluation of XAI methods and evaluation metrics for MIMIC-III mortality prediction, comparing the accuracy and cumulative preservation difference.

| Method | Acc (10%) ↓ | CPD ($K = 50$) ↑ |
|---|---|---|
| Extrmask | $0.930_{\pm0.005}$ | $0.204_{\pm0.007}$ |
| ContraLSP | $0.981_{\pm0.003}$ | $0.013_{\pm0.001}$ |
| TimeX++ | $0.991_{\pm0.001}$ | $0.027_{\pm0.002}$ |
| IG (Unsigned) | $0.974_{\pm0.001}$ | $\underline{0.342_{\pm0.021}}$ |
| IG (Signed) | $\mathbf{0.855_{\pm0.011}}$ | $0.248_{\pm0.010}$ |
| TIMING | $0.975_{\pm0.001}$ | $\mathbf{0.366_{\pm0.021}}$ |

### 3.2. New Metrics: Cumulative Prediction Difference and Preservation

To address the aforementioned limitations, we introduce two novel evaluation metrics—Cumulative Prediction Difference (CPD) and Cumulative Prediction Preservation (CPP)—designed to assess both directed and undirected attribution methods on an equal footing. Specifically, CPD sequentially removes the points with the highest absolute attributions and cumulatively measures the prediction differences between consecutive steps, where larger values indicate better performance. Formally, CPD is defined as:

$$\text{CPD}(x) = \sum_{k=0}^{K-1} \left\| F(x_k^\uparrow) - F(x_{k+1}^\uparrow) \right\|_1,$$

where $F(x) = (F_1(x), F_2(x), \ldots, F_C(x))$ is the model output probability vector for the input $x$, and $\|\cdot\|_1$ denotes the $\ell_1$ norm. CPD computes the cumulative difference between class probability vectors for consecutive inputs across all time steps, capturing the overall impact of perturbations on model predictions.

In a similar vein, CPP sequentially removes points with the lowest absolute attributions and measures prediction differences between consecutive steps. The CPP metric is defined as:

$$\text{CPP}(x) = \sum_{k=0}^{K-1} \left\| F(x_k^\downarrow) - F(x_{k+1}^\downarrow) \right\|_1,$$

where $x_k^\downarrow$ refers to the input after the removal of the top-$k$ points with the lowest absolute attributions. The smaller CPP indicates that the model's predictions remain stable when removing points deemed less important by the attribution method.

Our CPD and CPP metrics offer several compelling advantages. First, they are specifically designed to optimize the evaluation of an XAI method's faithfulness, providing a robust measure of attribution quality. Second, these metrics enable a fair comparison between signed and unsigned attribution methods and extend naturally to domains beyond the time series domain. However, for signed methods, alternating positive and negative points by absolute value order can

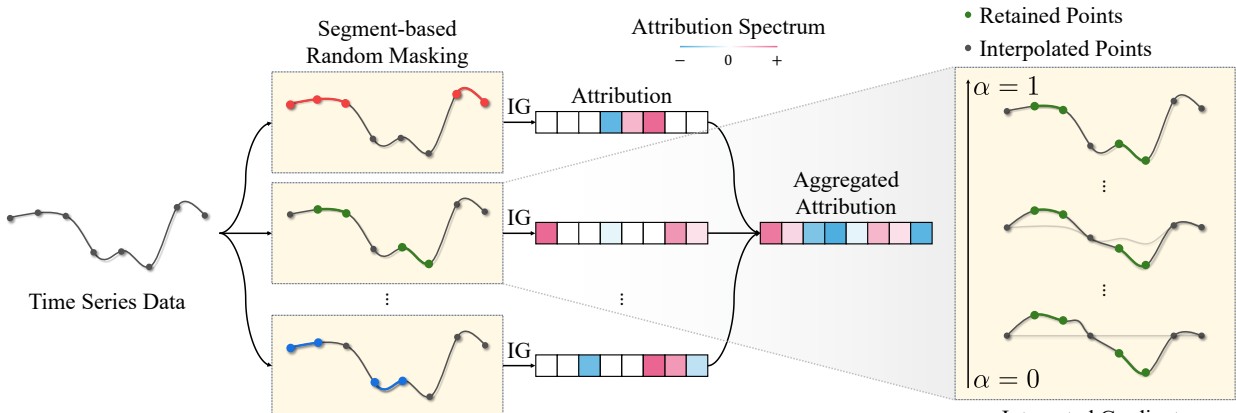

**Figure 2:** Overview of the Temporality-Aware Integrated Gradients (TIMING) framework for improved attribution in time series data. TIMING extends the traditional Integrated Gradients (IG) (Section 4.1) by incorporating temporal dependencies through segment-based random masking to handle disruptions in temporal relationships (Section 4.2). Our framework applies a randomization strategy to compute IG under varied conditions and aggregates the results to yield more robust feature attributions.

inflate the metric; they should avoid using attribution signs to manipulate point ordering. Last, by varying $K$ when computing CPD or CPP, these metrics provide a holistic evaluation of the overall ranking of attributions, offering deeper insights into their effectiveness.

## 4. Proposed Method

In this section, we first briefly review Integrated Gradients (IG) (Sundararajan et al., 2017) and discuss its limitations when applied to time series data. We then describe our novel method, Temporality-Aware Integrated Gradients (TIMING), along with its theoretical properties. The overall framework of our method is illustrated in Figure 2 and the detailed algorithm is provided in Appendix A.

### 4.1. Background: Integrated Gradients

We begin by reviewing the methodology of IG (Sundararajan et al., 2017). Formally, let $F : \mathbb{R}^{T \times D} \to [0, 1]^C$ be a time series classifier, where $T$ is the number of time steps, $D$ is the feature dimension, and $C$ is the number of classes. The classifier outputs class probabilities $F(x) = (F_1(x), \dots, F_C(x))$, satisfying $F_c(x) \geq 0$ and $\sum_{c=1}^{C} F_c(x) = 1$. The predicted class is given by $\hat{y} = \arg\max_{c \in \{1,\dots,C\}} F_c(x)$. IG computes point-wise attribution by integrating gradients along the straight-line path from a baseline $x'$ to the input $x$. In time series applications, the baseline $x'$ is typically chosen as 0. Thus, the formulation for time $t$ and dimension $d$ is as follows:

$$\text{IG}_{t,d}(x) = (x_{t,d} - x'_{t,d}) \int_{\alpha=0}^{1} \frac{\partial F_{\hat{y}}(x' + \alpha(x - x'))}{\partial x_{t,d}} \, d\alpha$$

$$= x_{t,d} \times \int_{\alpha=0}^{1} \frac{\partial F_{\hat{y}}(\alpha x)}{\partial x_{t,d}} \, d\alpha.$$

### 4.2. TIMING: Temporality-Aware Integrated Gradients

**Suboptimality of naive IG on time series data.** Directly applying IG in Section 4.1 to time series data introduces several non-trivial problems. First, when using a zero baseline, all points along the path in the time series are simply scaled down, and gradients are computed under this condition. This approach only captures changes when temporal relationships *remain consistent* and thereby fails to observe effects when temporal patterns are lost. In other words, while computing the impact of the current value, IG maintains the relationship with past and future values, making it difficult to interpret scenarios where such relationships break down. Another critical issue in time series data is the Out-of-Distribution (OOD) problem. When all points along the path are scaled down, the intermediate points may lie in OOD regions that the model has never encountered during training. In such cases, the gradients computed along the path may not contribute meaningfully to determining the importance, leading to unreliable attributions.

**Randomly retaining strategy.** To address these two issues, we first consider partially retaining certain points in the data when applying IG. By preserving some original values, we can observe how $F(x)$ changes when certain temporal relationships are disrupted. Specifically, we modify the zero baseline to $(\mathbf{1} - M) \odot x$, where $M \in \{0, 1\}^{T \times D}$ is a binary mask indicating which points are scaled down to zero ($M_{t,d} = 1$) versus retained ($M_{t,d} = 0$). In this way, each intermediate point remains closer to $x$, which helps mitigate the OOD problem. Concretely, intermediate points in the IG path can be represented as follows:

$$x' + \alpha(x - x') = \alpha(M \odot x) + (\mathbf{1} - M) \odot x.$$

Next, we define this partially masked version of IG, called MaskingIG, as:

$$\text{MaskingIG}_{t,d}(x, M) =$$
$$x_{t,d} M_{t,d} \times \int_{\alpha=0}^{1} \frac{\partial F_{\hat{y}}(\alpha(M \odot x) + (\mathbf{1} - M) \odot x)}{\partial x_{t,d}} \ d\alpha,$$

where $\odot$ denotes element-wise multiplication. To estimate the attribution of each point $x_{t,d}$, we then propose RandIG, which runs MaskingIG multiple times with random masks. By computing the expectation of IG under these random masks, we can obtain feature attribution $\forall x_{t,d}$:

$$\text{RandIG}_{t,d}(x; p) =$$
$$\mathbb{E}_{M_p} \left[ \text{MaskingIG}_{t,d}(x, M_p) \,\big|\, (M_p)_{t,d} = 1 \right],$$

where $M_p \in \{0, 1\}^{T \times D}$ and $(M_p)_{i,j} \sim \text{Bernoulli}(p)$ are independent $\forall 1 \le i \le T, 1 \le j \le D$.

**Combination of segments.** In RandIG, each point is randomly masked with probability $p$ *independently*. However, this approach can be suboptimal for time series data, as the retained points may not align with meaningful temporal structures. Time series data inherently exhibit temporal dependencies, where both individual points and subsequences carry rich semantic information (Leung et al., 2023). To better suit time series data, we propose segment-based masking instead of random point masking. Retaining several segments allows the model to preserve segment-level information, mitigating the OOD issue and enabling better consideration of scenarios where temporal relationships are either preserved or disrupted. Using the *Combination of Segments* strategy, we introduce *Temporality-Aware Integrated Gradients* (TIMING), formalized as:

$$\text{TIMING}_{t,d}(x; n, s_{min}, s_{max}) =$$
$$\mathbb{E}_{M \sim G(n, s_{min}, s_{max})} \left[ \text{MaskingIG}_{t,d}(x, M) \,\big|\, M_{t,d} = 1 \right],$$

where $G(n, s_{min}, s_{max})$ is a mask generator that selects $n$ segments of lengths within $[s_{min}, s_{max}]$. Instead of masking individual points, TIMING applies masking at the segment level, thereby reflecting temporal dependencies in each IG computation.

### 4.3. Theoretical Properties

**Applying randomness in the path.** Computing IG multiple times can yield accurate attributions but is often impractical due to its prohibitive computational cost. Instead, we propose a more cost-efficient approach that leverages randomization along the path. Formally, we introduce the *Effectiveness* of TIMING.

**Proposition 4.1** (Effectiveness). *Let $x, x' \in \mathbb{R}^{T \times D}$ be any input and baseline respectively, and let $M \in \{0, 1\}^{T \times D}$ be*

*a binary mask. Define the retained baseline combined with the input as:*

$$\tilde{x}(M) = (\mathbf{1} - M) \odot x + M \odot x',$$

*and consider the intermediate point in the path from $\tilde{x}(M)$ to $x$:*

$$z(\alpha; M) = \tilde{x}(M) + \alpha(x - \tilde{x}(M)), \quad \alpha \in [0, 1].$$

*Suppose the partial derivatives of the model output $F_{\hat{y}}$ are bounded along all of these paths. Then*

$$\int_0^1 \left| \frac{\partial F_{\hat{y}}(z(\alpha; M))}{\partial x_{t,d}} \right| \ d\alpha < \infty, \quad \forall \alpha \in [0, 1], \ t, \ d, \ M.$$

*Especially if $x' = 0$ and $M$ follows some probability distribution,*

$$\mathbb{E}_M \left[ MaskingIG_{t,d}(x, M) \,\big|\, M_{t,d} = 1 \right]$$
$$= x_{t,d} \times \int_{\alpha=0}^{1} \mathbb{E}_M \left[ \frac{\partial F_{\hat{y}}(z(\alpha; M))}{\partial x_{t,d}} \,\bigg|\, M_{t,d} = 1 \right] \ d\alpha.$$

Proposition 4.1 shows that we no longer need to compute IG repeatedly over different baselines. Instead, we can randomly select *one binary mask for each intermediate point in the IG path*, creating a highly fluctuating path from the baseline 0 to the input. The detailed proof of Proposition 4.1 is provided in Appendix B. In all of the following experiments, we adopt an efficient formulation of TIMING using a *single sample* to approximate the inter-expectation.

**Axiomatic properties.** TIMING satisfies several key axiomatic properties, ensuring its theoretical soundness and consistency with the original Integrated Gradients (IG) method (Sundararajan et al., 2017).

**Proposition 4.2** (Sensitivity). *Let $x$ and $x'$ be two inputs that differ in exactly one point $(t, d)$, such that $x_{t,d} \ne x'_{t,d}$ and $x_{i,j} = x'_{i,j}$ for all $(i, j) \ne (t, d)$. If $F(x) \ne F(x')$, then $TIMING_{t,d}(x) \ne 0$.*

**Proposition 4.3** (Implementation Invariance). *TIMING maintains consistency across functionally equivalent models, ensuring identical attributions if two models produce identical outputs for all inputs.*

These properties are practically important since they ensure that TIMING provides reliable and interpretable feature attributions that are consistent, sensitive to changes, The detailed proof for these propositions is in Appendix B.

**Proposition 4.4** (Incompleteness). *Let $x$ be an input and $x'$ be a baseline. Then, the sum of the attributions $\{TIMING_{t,d}(x)\}$ assigned by our method across all features is not guaranteed to equal the difference in model outputs. Hence, our method does not satisfy the* completeness *axiom as defined for IG.*

**Table 2:** Performance comparison of various XAI methods on MIMIC-III mortality prediction with zero substitution. Results are aggregated with mean ± standard error over five random cross-validation repetitions and presented for both 20% masking and cumulative masking strategies, with multiple metrics including cumulative prediction difference (CPD) for two values of $K = 50, 100$, accuracy (Acc), cross-entropy (CE), sufficiency (Suff), and comprehensiveness (Comp).

| Method | Cumulative Masking | | 20% Masking | | | |
|---|---|---|---|---|---|---|
| | CPD ($K = 50$) ↑ | CPD ($K = 100$) ↑ | Acc ↓ | CE ↑ | Suff $* 10^2$ ↓ | Comp $* 10^2$ ↑ |
| FO | $0.016_{\pm0.002}$ | $0.034_{\pm0.004}$ | $0.991_{\pm0.001}$ | $0.101_{\pm0.006}$ | $1.616_{\pm0.531}$ | $-0.258_{\pm0.180}$ |
| AFO | $0.120_{\pm0.008}$ | $0.177_{\pm0.013}$ | $0.975_{\pm0.002}$ | $0.121_{\pm0.007}$ | $1.484_{\pm0.306}$ | $-0.698_{\pm0.257}$ |
| GradSHAP | $0.327_{\pm0.021}$ | $0.447_{\pm0.030}$ | $0.975_{\pm0.002}$ | $0.136_{\pm0.008}$ | $0.253_{\pm0.217}$ | $0.570_{\pm0.536}$ |
| DeepLIFT | $0.142_{\pm0.010}$ | $0.189_{\pm0.014}$ | $0.974_{\pm0.002}$ | $\underline{0.374}_{\pm0.005}$ | $0.325_{\pm0.076}$ | $-0.001_{\pm0.176}$ |
| LIME | $0.071_{\pm0.004}$ | $0.087_{\pm0.005}$ | $0.988_{\pm0.001}$ | $0.103_{\pm0.008}$ | $-1.875_{\pm0.081}$ | $-0.259_{\pm0.257}$ |
| FIT | $0.015_{\pm0.001}$ | $0.032_{\pm0.002}$ | $0.991_{\pm0.001}$ | $0.103_{\pm0.006}$ | $1.620_{\pm0.686}$ | $0.008_{\pm0.119}$ |
| WinIT | $0.020_{\pm0.001}$ | $0.038_{\pm0.002}$ | $0.989_{\pm0.001}$ | $0.106_{\pm0.006}$ | $1.261_{\pm0.658}$ | $0.250_{\pm0.147}$ |
| Dynamask | $0.052_{\pm0.002}$ | $0.079_{\pm0.004}$ | $0.974_{\pm0.002}$ | $0.131_{\pm0.008}$ | $0.081_{\pm0.374}$ | $1.626_{\pm0.218}$ |
| Extrmask | $0.204_{\pm0.007}$ | $0.281_{\pm0.009}$ | $\underline{0.932}_{\pm0.005}$ | $\mathbf{0.485}_{\pm0.022}$ | $\mathbf{-8.434}_{\pm0.382}$ | $\mathbf{23.370}_{\pm1.088}$ |
| ContraLSP | $0.013_{\pm0.001}$ | $0.028_{\pm0.002}$ | $\mathbf{0.921}_{\pm0.006}$ | $0.301_{\pm0.013}$ | $\underline{-7.114}_{\pm0.306}$ | $12.690_{\pm0.998}$ |
| TimeX | $0.064_{\pm0.007}$ | $0.101_{\pm0.009}$ | $0.974_{\pm0.002}$ | $0.117_{\pm0.003}$ | $3.810_{\pm0.560}$ | $-1.701_{\pm0.166}$ |
| TimeX++ | $0.027_{\pm0.002}$ | $0.051_{\pm0.004}$ | $0.987_{\pm0.001}$ | $0.095_{\pm0.005}$ | $1.885_{\pm0.328}$ | $-0.936_{\pm0.127}$ |
| IG | $\underline{0.342}_{\pm0.021}$ | $0.469_{\pm0.030}$ | $0.974_{\pm0.001}$ | $0.132_{\pm0.008}$ | $0.403_{\pm0.156}$ | $0.118_{\pm0.561}$ |
| TIMING | $\mathbf{0.366}_{\pm0.021}$ | $\mathbf{0.505}_{\pm0.029}$ | $0.975_{\pm0.002}$ | $0.136_{\pm0.008}$ | $0.242_{\pm0.136}$ | $0.436_{\pm0.562}$ |

While standard IG ensures that all attributions sum to the overall difference in model output for one baseline, our method broadens the interpretation by examining multiple baseline contexts. This broader perspective can offer richer insights into when and how each feature contributes across different masking or baseline conditions. However, this flexibility inevitably sacrifices the original completeness property that IG guarantees.

## 5. Experiments

This section presents a comprehensive evaluation of the empirical effectiveness of TIMING. We begin by describing the experimental setup in Section 5.1, followed by answering the key research questions:

- Can TIMING faithfully capture the points that truly influence the model's predictions? (Section 5.2)
- Do the individual components of TIMING truly contribute to capturing model explanations? (Section 5.3)
- Are the explanations provided by TIMING practically meaningful and interpretable for end-users? (Section 5.4)
- How practical is TIMING in terms of hyperparameter sensitivity and time complexity? (Section 5.4)

### 5.1. Experimental Setup

**Datasets.** Following existing state-of-the-art literature (Liu et al., 2024b;a), we evaluate TIMING on both synthetic and real-world datasets. For synthetic datasets, we utilize Switch-Feature (Tonekaboni et al., 2020; Liu et al., 2024b) and State (Tonekaboni et al., 2020; Crabbé & Van Der Schaar, 2021). For real-world datasets, we

employ MIMIC-III (Johnson et al., 2016), Personal Activity Monitoring (PAM) (Reiss & Stricker, 2012), Boiler (Shohet et al., 2019), Epilepsy (Andrzejak et al., 2001), Wafer (Dau et al., 2019), and Freezer (Dau et al., 2019). These datasets span a wide range of real-world time series domains, ensuring a comprehensive evaluation of TIMING's effectiveness. Detailed descriptions of the datasets are provided in Appendix D.

**XAI baselines.** We conduct a comprehensive comparison of TIMING against 13 baseline methods: FO (Suresh et al., 2017), AFO (Tonekaboni et al., 2020), IG (Sundararajan et al., 2017), GradSHAP (Lundberg & Lee, 2017), DeepLIFT (Shrikumar et al., 2017), LIME (Ribeiro et al., 2016), FIT (Tonekaboni et al., 2020), WinIT (Leung et al., 2023), Dynamask (Crabbé & Van Der Schaar, 2021), Extrmask (Enguehard, 2023), ContraLSP (Liu et al., 2024b), TimeX (Queen et al., 2024), and TimeX++ (Liu et al., 2024a). These baselines encompass a diverse range of approaches, including modality-agnostic XAI methods—FO, AFO, IG, GradSHAP, DeepLIFT, LIME—and time series-specific XAI techniques such as FIT, WinIT, Dynamask, Extrmask, ContraLSP, TimeX, and TimeX++, ensuring a robust evaluation of TIMING.

**Model architectures.** We primarily evaluate TIMING on black-box classifiers using a single-layer GRU (Chung et al., 2014), following the experimental protocols of prior works (Tonekaboni et al., 2020; Leung et al., 2023; Crabbé & Van Der Schaar, 2021; Enguehard, 2023; Liu et al., 2024b;a). To demonstrate the model-agnostic versatility of TIMING, we assess its performance on CNNs (Krizhevsky et al., 2012) and Transformers (Vaswani et al., 2017) in Appendix E.

**Table 3:** Performance comparison of various XAI methods on real-world datasets with 10% feature masking. Results are aggregated as mean ± standard error over five random cross-validation repetitions and presented across multiple datasets, including MIMIC-III, PAM, Boiler (Multivariate), Epilepsy, Wafer, and Freezer (Univariate). Evaluation metrics include cumulative prediction difference (CPD) attribution performance under two feature substitution strategies: average substitution (Avg.) and zero substitution (Zero).

| | MIMIC-III | | PAM | | Boiler | | Epilepsy | | Wafer | | Freezer | |
| Method | Avg. | Zero | Avg. | Zero | Avg. | Zero | Avg. | Zero | Avg. | Zero | Avg. | Zero |
|---|---|---|---|---|---|---|---|---|---|---|---|---|
| AFO | $0.127_{\pm 0.009}$ | $0.227_{\pm 0.017}$ | $0.140_{\pm 0.009}$ | $0.200_{\pm 0.013}$ | $0.262_{\pm 0.020}$ | $0.349_{\pm 0.035}$ | $0.028_{\pm 0.003}$ | $0.030_{\pm 0.004}$ | $0.018_{\pm 0.003}$ | $0.018_{\pm 0.003}$ | $0.143_{\pm 0.054}$ | $0.143_{\pm 0.054}$ |
| GradSHAP | $\mathbf{0.250}_{\pm 0.015}$ | $0.522_{\pm 0.038}$ | $0.421_{\pm 0.014}$ | $0.518_{\pm 0.012}$ | $0.752_{\pm 0.055}$ | $0.747_{\pm 0.092}$ | $\underline{0.052}_{\pm 0.004}$ | $\underline{0.054}_{\pm 0.004}$ | $0.485_{\pm 0.014}$ | $0.485_{\pm 0.014}$ | $0.397_{\pm 0.110}$ | $0.397_{\pm 0.110}$ |
| Extrmask | $0.154_{\pm 0.008}$ | $0.305_{\pm 0.010}$ | $0.291_{\pm 0.007}$ | $0.380_{\pm 0.009}$ | $0.338_{\pm 0.028}$ | $0.400_{\pm 0.031}$ | $0.028_{\pm 0.003}$ | $0.029_{\pm 0.003}$ | $0.202_{\pm 0.026}$ | $0.202_{\pm 0.026}$ | $0.176_{\pm 0.057}$ | $0.176_{\pm 0.057}$ |
| ContraLSP | $0.048_{\pm 0.003}$ | $0.051_{\pm 0.004}$ | $0.046_{\pm 0.007}$ | $0.059_{\pm 0.011}$ | $0.408_{\pm 0.035}$ | $0.496_{\pm 0.043}$ | $0.016_{\pm 0.001}$ | $0.016_{\pm 0.001}$ | $0.121_{\pm 0.032}$ | $0.121_{\pm 0.032}$ | $0.176_{\pm 0.055}$ | $0.176_{\pm 0.055}$ |
| TimeX++ | $0.017_{\pm 0.002}$ | $0.074_{\pm 0.006}$ | $0.057_{\pm 0.004}$ | $0.070_{\pm 0.004}$ | $0.124_{\pm 0.028}$ | $0.208_{\pm 0.043}$ | $0.030_{\pm 0.004}$ | $0.032_{\pm 0.004}$ | $0.000_{\pm 0.000}$ | $0.000_{\pm 0.000}$ | $0.216_{\pm 0.056}$ | $0.216_{\pm 0.056}$ |
| IG | $\underline{0.243}_{\pm 0.015}$ | $\underline{0.549}_{\pm 0.039}$ | $\underline{0.448}_{\pm 0.013}$ | $\underline{0.573}_{\pm 0.022}$ | $\underline{0.759}_{\pm 0.053}$ | $\underline{0.752}_{\pm 0.013}$ | $\underline{0.052}_{\pm 0.004}$ | $\underline{0.054}_{\pm 0.004}$ | $\underline{0.500}_{\pm 0.017}$ | $\underline{0.500}_{\pm 0.017}$ | $\underline{0.405}_{\pm 0.111}$ | $\underline{0.405}_{\pm 0.111}$ |
| TIMING | $\mathbf{0.250}_{\pm 0.015}$ | $\mathbf{0.597}_{\pm 0.037}$ | $\mathbf{0.463}_{\pm 0.007}$ | $\mathbf{0.602}_{\pm 0.033}$ | $\mathbf{1.259}_{\pm 0.065}$ | $\mathbf{1.578}_{\pm 0.085}$ | $\mathbf{0.057}_{\pm 0.005}$ | $\mathbf{0.060}_{\pm 0.005}$ | $\mathbf{0.674}_{\pm 0.014}$ | $\mathbf{0.674}_{\pm 0.014}$ | $\mathbf{0.409}_{\pm 0.109}$ | $\mathbf{0.409}_{\pm 0.109}$ |

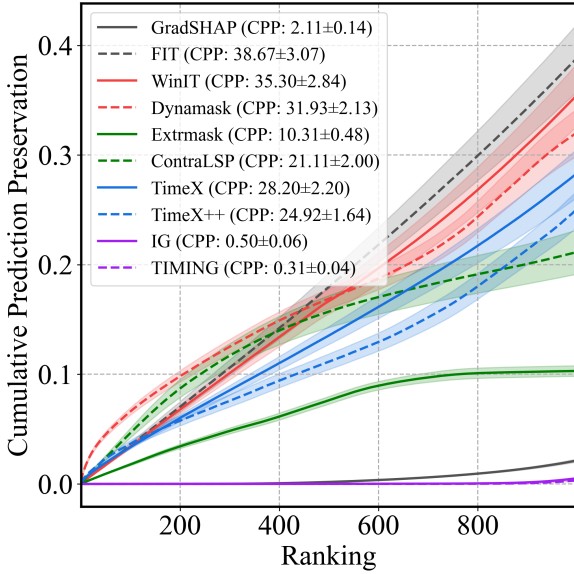

**Figure 3:** Cumulative Prediction Preservation (CPP) comparison of XAI methods on MIMIC-III mortality prediction with zero substitution. Results are averaged over five cross-validation runs, with 10% random masking of observed points alongside all missing points.

**Evaluation metrics.** As we propose new evaluation metrics for time series XAI—CPD and CPP—and address the limitations of existing metrics, we primarily employ these metrics on both synthetic and real-world datasets. Nevertheless, we also report results based on established metrics: AUP and AUR for synthetic datasets (Liu et al., 2024a), as well as accuracy, cross-entropy, sufficiency, and comprehensiveness for real-world datasets (Liu et al., 2024b). Detailed explanations of these metrics are provided in Appendix C.

**Implementation details.** In all tables, the best and second-best performance values are shown in **bold** and underlined, respectively. Further implementation details are available at https://github.com/drumpt/TIMING.

## 5.2. Main Results

**Result on real-world datasets.** As shown in Table 2, TIMING achieves the best performance, with average Cumulative Prediction Difference (CPD) scores of 0.366 ($K = 50$) and 0.505 ($K = 100$). Additionally, modality-agnostic gradient-based methods, such as IG and Grad-SHAP, demonstrate strong capabilities in identifying important features. Recent masking-based methods, including Extrmask and ContraLSP, achieve state-of-the-art performance based on standard evaluation metrics. However, as illustrated in Figure 3, these methods exhibit a critical limitation—unimportant points can significantly affect model predictions. Our analysis of CPD and CPP scores reveals that these masking-based methods tend to assign zero importance to negatively important features, potentially overlooking their impact on model behavior. To further validate our findings, we extended our evaluation to additional real-world datasets. Also demonstrated in Table 3, TIMING consistently achieves state-of-the-art performance across all datasets. Specifically, TIMING improves zero substitution CPD performance with relative increases of 8.7% for MIMIC-III, 5.1% for PAM, 109.8% for Boiler, 11.1% for Epilepsy, 34.8% for Wafer, 1.0% for Freezer. These results highlight the robustness of TIMING in diverse real-world scenarios.

**Result on synthetic datasets.** Although real-world datasets are more crucial for practical applications, we also conducted experiments on the Switch-Feature and State datasets following Liu et al. (2024b). As illustrated in Table 4, ContraLSP and Extrmask outperform other methods in estimating the true saliency map on the Switch-Feature dataset. However, this superior performance in saliency map estimation comes at the cost of sacrificing aspects of model explanations, as reflected by their CPD scores. In contrast, while TIMING may not excel in estimating true saliency maps, it provides robustly effective model explanations, achieving high CPD scores on both synthetic datasets. This demonstrates that TIMING not only excels in real-world scenarios but also maintains strong model explanation capabilities in controlled synthetic environments.

**Table 4:** Performance comparison of various XAI methods on Switch Feature and State datasets. Results are reported as mean ± standard error over five cross-validation repetitions, evaluated using AUP, AUR, and CPD (10% masking) for true saliency map and cumulative masking strategies.

| Method | Switch-Feature | | |
| | CPD ↑ | AUP ↑ | AUR ↑ |
|---|---|---|---|
| FO | $0.191_{\pm 0.006}$ | $0.902_{\pm 0.009}$ | $0.374_{\pm 0.006}$ |
| AFO | $0.182_{\pm 0.007}$ | $0.836_{\pm 0.012}$ | $0.416_{\pm 0.008}$ |
| GradSHAP | $0.196_{\pm 0.006}$ | $0.892_{\pm 0.010}$ | $0.387_{\pm 0.006}$ |
| DeepLIFT | $0.196_{\pm 0.007}$ | $0.918_{\pm 0.019}$ | $0.432_{\pm 0.011}$ |
| LIME | $0.195_{\pm 0.006}$ | $0.949_{\pm 0.015}$ | $0.391_{\pm 0.016}$ |
| FIT | $0.106_{\pm 0.001}$ | $0.522_{\pm 0.005}$ | $0.437_{\pm 0.002}$ |
| Dynamask | $0.069_{\pm 0.001}$ | $0.362_{\pm 0.003}$ | $0.754_{\pm 0.008}$ |
| Extrmask | $0.174_{\pm 0.002}$ | $\mathbf{0.978}_{\pm \mathbf{0.004}}$ | $0.745_{\pm 0.007}$ |
| ContraLSP | $0.158_{\pm 0.002}$ | $0.970_{\pm 0.005}$ | $\mathbf{0.851}_{\pm \mathbf{0.005}}$ |
| IG | $0.196_{\pm 0.007}$ | $0.918_{\pm 0.019}$ | $0.433_{\pm 0.011}$ |
| TIMING | $\mathbf{0.208}_{\pm \mathbf{0.003}}$ | $0.926_{\pm 0.011}$ | $0.434_{\pm 0.015}$ |

| Method | State | | |
| | CPD ↑ | AUP ↑ | AUR ↑ |
|---|---|---|---|
| FO | $0.158_{\pm 0.004}$ | $0.882_{\pm 0.021}$ | $0.303_{\pm 0.005}$ |
| AFO | $0.143_{\pm 0.007}$ | $0.809_{\pm 0.037}$ | $0.374_{\pm 0.007}$ |
| GradSHAP | $0.156_{\pm 0.004}$ | $0.857_{\pm 0.019}$ | $0.315_{\pm 0.009}$ |
| DeepLIFT | $0.162_{\pm 0.002}$ | $0.926_{\pm 0.008}$ | $0.359_{\pm 0.008}$ |
| LIME | $\mathbf{0.163}_{\pm \mathbf{0.002}}$ | $\mathbf{0.944}_{\pm \mathbf{0.008}}$ | $0.333_{\pm 0.010}$ |
| FIT | $0.057_{\pm 0.000}$ | $0.483_{\pm 0.001}$ | $\mathbf{0.607}_{\pm \mathbf{0.002}}$ |
| Dynamask | $0.052_{\pm 0.001}$ | $0.335_{\pm 0.003}$ | $0.506_{\pm 0.002}$ |
| Extrmask | $0.055_{\pm 0.001}$ | $0.557_{\pm 0.024}$ | $0.012_{\pm 0.001}$ |
| ContraLSP | $0.025_{\pm 0.000}$ | $0.495_{\pm 0.011}$ | $0.015_{\pm 0.001}$ |
| IG | $0.162_{\pm 0.002}$ | $0.922_{\pm 0.009}$ | $0.357_{\pm 0.008}$ |
| TIMING | $\mathbf{0.163}_{\pm \mathbf{0.002}}$ | $0.921_{\pm 0.010}$ | $0.355_{\pm 0.008}$ |

**Table 5:** Ablation study comparing TIMING's segment-based masking strategy against IG and a random masking IG (RandIG). We report CPD with $K = 50$ on the MIMIC-III benchmark under both average and zero substitutions.

| Method | Avg. | Zero |
|---|---|---|
| IG | $0.172_{\pm 0.011}$ | $0.342_{\pm 0.021}$ |
| RandIG ($p = 0.3$) | $0.175_{\pm 0.011}$ | $0.350_{\pm 0.022}$ |
| RandIG ($p = 0.5$) | $0.175_{\pm 0.011}$ | $0.353_{\pm 0.022}$ |
| RandIG ($p = 0.7$) | $0.174_{\pm 0.011}$ | $0.354_{\pm 0.022}$ |
| TIMING | $\mathbf{0.177}_{\pm \mathbf{0.011}}$ | $\mathbf{0.366}_{\pm \mathbf{0.021}}$ |

### 5.3. Ablation Study

We perform an ablation study on TIMING to assess the impact of each component. As TIMING builds on Integrated Gradients (IG) with segment-based masking, we compare it to standard IG and RandIG. As shown in Table 5, TIMING outperforms both methods across substitution strategies, with a particularly large gain in the Zero substitution settings. This result shows RandIG's limitation in solely disrupting temporal dependencies. In contrast, TIMING preserves structured information by leveraging segment-level temporal patterns. This improves OOD generalization and reinforces the need for segment-based attribution in time series explainability.

### 5.4. Further Analysis

**Qualitative analysis.** We qualitatively assess TIMING for coherence—examining whether its explanations are meaningful and interpretable. In the MIMIC-III mortality benchmark, Figures 6 and 7 (true positives) highlight feature index 9 (lactate) as most salient, consistent with clinical knowledge that *elevated lactate contributes to lactic acidosis, strongly linked to mortality* (Villar et al., 2019; Bernhard et al., 2020). Conversely, Figures 8 and 9 (true negatives) align with known patterns: a *low BUN/Cr ratio does not indicate mortality risk* (Tanaka et al., 2017; Ma et al., 2023), and *lower systolic/diastolic blood pressures promote patient stability* (Group, 2015; Brunström & Carlberg, 2018). Further analyses in Figure 10 reveal that TIMING provides compact and reasonable feature attributions.

**Table 6:** Hyperparameter sensitivity analysis for $(n, s_{min}, s_{max})$, reporting CPD ($K = 50$) on MIMIC-III with average and zero substitutions.

| $(n, s_{min}, s_{max})$ | Avg. | Zero |
|---|---|---|
| $(10, 1, 10)$ | $0.173_{\pm 0.011}$ | $0.345_{\pm 0.021}$ |
| $(10, 1, 48)$ | $0.175_{\pm 0.011}$ | $0.354_{\pm 0.021}$ |
| $(10, 10, 10)$ | $0.173_{\pm 0.011}$ | $0.347_{\pm 0.021}$ |
| $(10, 10, 48)$ | $0.176_{\pm 0.011}$ | $0.356_{\pm 0.021}$ |
| $(100, 1, 10)$ | $0.175_{\pm 0.011}$ | $0.354_{\pm 0.021}$ |
| $(100, 1, 48)$ | $0.176_{\pm 0.011}$ | $0.365_{\pm 0.021}$ |
| $(100, 10, 10)$ | $0.175_{\pm 0.011}$ | $0.358_{\pm 0.021}$ |
| $(100, 10, 48)$ | $0.174_{\pm 0.011}$ | $0.363_{\pm 0.021}$ |
| $(50, 1, 10)$ | $0.174_{\pm 0.011}$ | $0.351_{\pm 0.021}$ |
| $(50, 1, 48)$ | $\mathbf{0.177}_{\pm \mathbf{0.011}}$ | $0.365_{\pm 0.021}$ |
| $(50, 10, 10)$ | $0.175_{\pm 0.011}$ | $0.355_{\pm 0.021}$ |
| TIMING $(50, 10, 48)$ | $\mathbf{0.177}_{\pm \mathbf{0.011}}$ | $\mathbf{0.366}_{\pm \mathbf{0.021}}$ |

**Hyperparameter sensitivity.** TIMING has three hyperparameters—$n$ for the number of filled segments, $s_{min}$ for minimum segment length, and $s_{max}$ for maximum segment length. As shown in Table 6, TIMING showcases strong robustness to hyperparameter choices, with a maximum discrepancy in the average performance of only 0.04 (Avg. substitution) and 0.019 (zero substitution) across all configurations. Moderate segment lengths ($s_{min} = 10$) paired with a larger maximum ($s_{max} = 48$) tend to perform best, while a small maximum ($s_{max} = 10$) slightly degrades performance. Meanwhile, our default setting of $(n, s_{min}, s_{max}) = (50, 10, 48)$ achieves optimal results by balancing these factors.

**Computational efficiency.** We further analyze the time complexity of TIMING and other baselines in Figure 4, focusing on methods that do not require post-training, as training-based algorithms inherently take much longer than their counterparts and are not a fair comparison. As shown in Figure 4, TIMING achieves a competitive complexity of

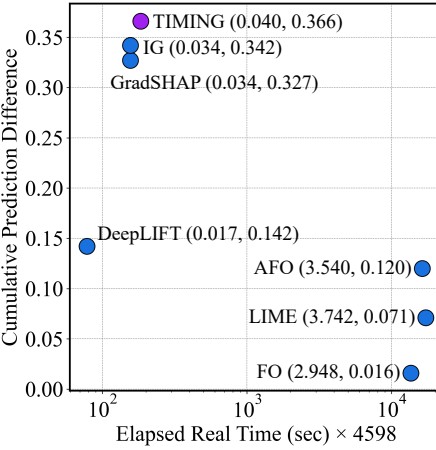

**Figure 4:** Computational efficiency analysis of TIMING and baselines. We report elapsed real time (sec) on a logarithmic scale for all test samples in the MIMIC-III benchmark, alongside CPD with $K = 50$. Each ordered pair $(x, \cdot)$ represents per-sample elapsed real-time as the $x$-coordinate.

0.04 sec/sample while delivering the best CPD performance. This highlights TIMING's optimal trade-off between efficiency and thoroughness.

## 6. Related Work

**Modality-agnostic explainable artificial intelligence methods.** Deep neural networks have become a dominant paradigm in machine learning, growing more complex over time. While achieving high accuracy, they often act as black boxes, offering little insight into how predictions are made (Christoph, 2020). This lack of transparency can undermine trust and accountability, especially in high-stakes applications (Rudin, 2019). Modality-agnostic explainable artificial intelligence (XAI) methods, such as SHAP (Lundberg & Lee, 2017), LIME (Ribeiro et al., 2016), Integrated Gradients (IG) (Sundararajan et al., 2017), and DeepLIFT (Shrikumar et al., 2017), aim to address this challenge by attributing model predictions to individual features. These methods help provide insight into how input features influence the output. Additionally, perturbation-based methods like Feature Occlusion (FO) (Suresh et al., 2017) and Augmented Feature Occlusion (AFO) (Tonekaboni et al., 2020) are commonly used to evaluate feature importance. FO works by masking individual feature values, replacing them with zeros or uniform noise, and observing the impact on the model's predictions. In contrast, AFO replaces features with the corresponding values from random training samples, rather than introducing noise. While these methods can help explain predictions, they are predominantly validated on the image and language domains (Selvaraju et al., 2017; Zhou et al., 2016; Ross et al., 2017; Danilevsky et al., 2020), and thus limited in the context of time series data, where the temporal dependencies between observations are crucial for accurate explanations.

**Explainable artificial intelligence for time series data.** Developing explainable artificial intelligence (XAI) methods for time series data encounters unique challenges due to the temporal dependencies inherent in the data. The relationships between data points are influenced by their order and historical context. Naively adopting conventional modality-agnostic XAI methods, which treat observations as independent, often fail to capture these dynamics. Recent methods address this by incorporating temporal dependencies. Specifically, FIT (Tonekaboni et al., 2020) quantifies feature importance by measuring shifts in the model's predictive distribution using KL divergence. WinIT (Leung et al., 2023) extends this by considering temporal dependencies between observations and aggregating importance over time. Dynamask (Crabbé & Van Der Schaar, 2021) learns dynamic masks for feature importance at each time step, while Extrmask (Enguehard, 2023) further learns perturbations to better capture temporal dynamics. ContraLSP (Liu et al., 2024b) tackles distribution shifts using contrastive learning with sparse gates. TimeX (Queen et al., 2024) trains interpretable surrogates to mimic pre-trained models, ensuring faithfulness while preserving temporal relationships. TimeX++ (Liu et al., 2024a) improves upon this with an information bottleneck framework to avoid trivial solutions and distribution shifts. Beyond these works, MIL-LET (Early et al., 2024) and TimeMIL (Chen et al., 2024) employ multiple-instance learning to provide explanations that operate only within the architectures they propose. Despite substantial advancements in XAI methods, critical gaps persist in capturing the directional influence of individual data points and developing a cohesive framework for evaluating their significance—challenges that this work directly tackles.

## 7. Conclusion

In this paper, we have proposed a time series XAI method by identifying critical limitations in existing attribution methods that fail to capture directional attributions and rely on flawed evaluation metrics. To this end, we introduced novel metrics—Cumulative Prediction Difference (CPD) and Cumulative Prediction Preservation (CPP)—to address these issues, revealing that classical Integrated Gradients (IG) outperforms recent methods. Building on this insight, we proposed TIMING, an enhanced integrated gradients approach that addresses the limitations of conventional IG with segment-based masking strategies, which effectively capture complex temporal dependencies while avoiding out-of-distribution samples. Extensive experiments demonstrated TIMING's superior performance in attribution faithfulness, coherence, and efficiency. We believe this work bridges the gap between model development and practical XAI in time series, offering reliable, interpretable insights for real-world applications.

## Acknowledgments

This work was supported by the Institute of Information & Communications Technology Planning & Evaluation (IITP) (No. RS-2019-II190075, Artificial Intelligence Graduate School Program (KAIST), No. 2022-0-00984, Development of Artificial Intelligence Technology for Personalized Plug-and-Play Explanation and Verification of Explanation) and the National Research Foundation of Korea (NRF) (No. RS-2023-00209060, A Study on Optimization and Network Interpretation Method for Large-Scale Machine Learning) grant funded by the Korean government (MSIT).

## Impact Statement

This paper presents work whose goal is to advance explainable AI for time series analysis. Our contributions enhance model transparency in safety-critical applications such as healthcare, energy systems, and infrastructure monitoring, enabling clinicians, engineers, and operators to better understand and trust AI-driven decisions. This work promotes responsible AI deployment by providing more reliable explanations that support human expertise and improve decision-making in domains where interpretability is crucial for safety and effectiveness.

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

# Appendix

# A. Algorithm

The detailed procedure for the efficient version of TIMING that we used in our experiments is provided in Algorithm 1.

---

**Algorithm 1** TIMING

---

**Input:** Test instance $x \in \mathbb{R}^{T \times D}$, Pre-trained time series classifier $F$, # of segments $n$, Minimum segment length $s_{min}$, Maximum segment length $s_{max}$, # of samples $n_{samples}$

$\alpha, \hat{y} \leftarrow 0, \arg\max_{c \in \{1,\ldots,C\}} F_c(x)$

Baselines, unmaskcount, TotalGrad $\leftarrow \mathbf{0}_{T \times D}, \mathbf{0}_{T \times D}, \mathbf{0}_{T \times D}$

**while** $\alpha < 1$ **do**

   // Parallelized in our implementation

   $\tilde{x} \leftarrow$ Baselines $+ \alpha (x -$ Baselines$)$

   **for** $i = 1$ to $n$ **do**

      // Parallelized in our implementation

      $l, d, s \leftarrow$ RandomSample$([s_{\min}, s_{\max}])$, RandomSample$(\{0, \ldots, D-1\})$, RandomSample$(\{0, \ldots, T-l\})$

      $e \leftarrow s + l$

      $\tilde{x}_{s:e,d} \leftarrow x_{s:e,d}$

      unmaskcount$_{s:e,d} \leftarrow$ unmaskcount$_{s:e,d} + 1$

   **end for**

   grad $\leftarrow \partial F_{\hat{y}}(\tilde{x}) / \partial \tilde{x}$

   grad$_{s:c,d} \leftarrow \mathbf{0}$

   TotalGrad $\leftarrow$ TotalGrad $+$ grad

   $\alpha \leftarrow \alpha + 1/n_{samples}$

**end while**

$\mathcal{A}(F, x) \leftarrow ($TotalGrad $\odot (x -$ Baselines$)) / (n_{samples} -$ unmaskcount$)$

**Output:** Attribution $\mathcal{A}(F, x)$

---

# B. Proof of Propositions

## B.1. Proof of Proposition 4.1

*Proof.* Since $x' = 0$, we have $\tilde{x}(M) = (\mathbf{1} - M) \odot x$. Therefore, the intermediate point $z(\alpha; M)$ on the path from $\tilde{x}(M)$ to $x$ is:

$$z(\alpha; M) = \tilde{x}(M) + \alpha(x - \tilde{x}(M)) = \alpha(M \odot x) + (\mathbf{1} - M) \odot x, \quad \forall \alpha \in [0, 1].$$

By bounded partial derivatives assumption, there exists $L > 0$ such that

$$\left| \frac{\partial F_{\hat{y}}(z(\alpha; M))}{\partial x_{t,d}} \right| \leq L \quad \forall \alpha \in [0, 1], \ t, \ d, \ M.$$

Hence,

$$\int_0^1 \left| \frac{\partial F_{\hat{y}}(z(\alpha; M))}{\partial x_{t,d}} \right| d\alpha \leq \int_0^1 L \, d\alpha = L < \infty.$$

In particular, for each $(t, d)$,

$$\left| x_{t,d} \times \int_{\alpha=0}^1 \left| \frac{\partial F_{\hat{y}}(z(\alpha; M))}{\partial x_{t,d}} \right| d\alpha \right| \leq L |x_{t,d}|.$$

Since $M$ is drawn from a probability distribution and we assume $\Pr(M_{t,d} = 1) > 0$, the event $\{M_{t,d} = 1\}$ occurs with nonzero probability, and thus $\mathbb{E}[\cdot \mid M_{t,d} = 1]$ is well-defined. Therefore,

$$\mathbb{E}_M \left[ x_{t,d} M_{t,d} \int_0^1 \left| \frac{\partial F_{\hat{y}}(z(\alpha; M))}{\partial x_{t,d}} \right| d\alpha \ \middle| \ M_{t,d} = 1 \right]$$

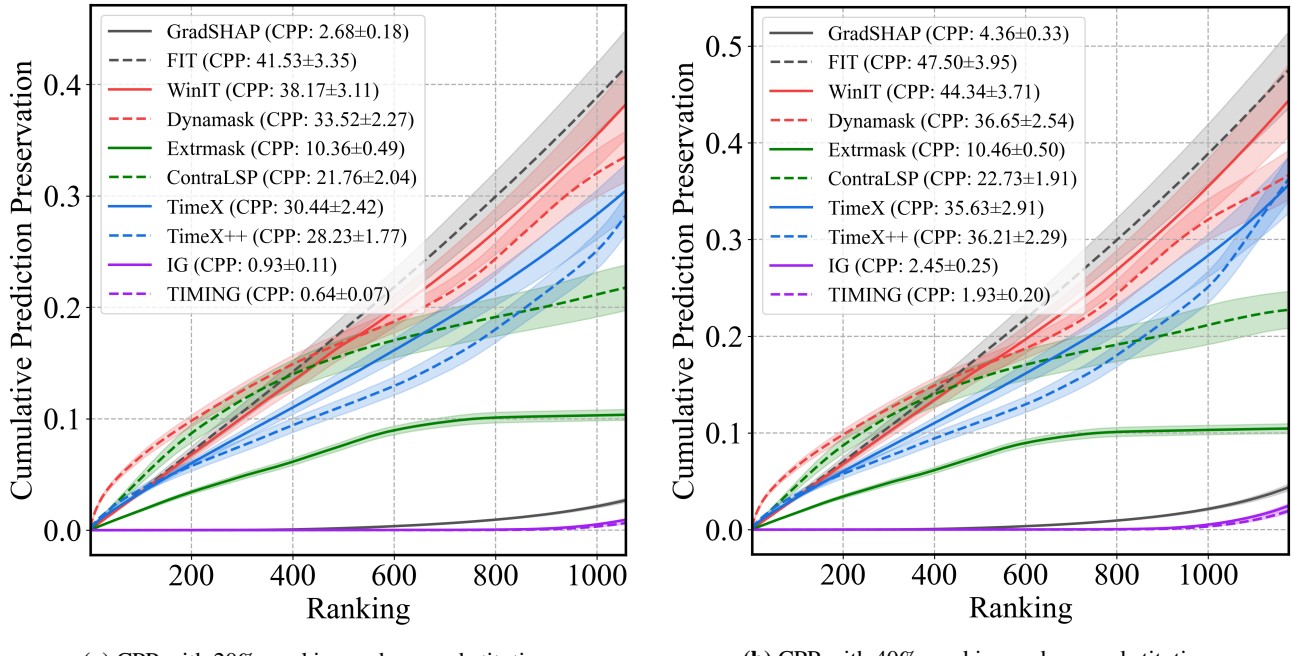

**(a)** CPP with 20% masking and zero substitution.

**(b)** CPP with 40% masking and zero substitution.

**Figure 5:** Cumulative Prediction Preservation (CPP) comparison of XAI methods on MIMIC-III mortality prediction with zero substitution. Results are averaged over five cross-validation runs, with 20% and 40% random masking of observed points alongside all missing points.

is finite. By the Fubini-Tonelli theorem, we may interchange the integral and the expectation. Hence, we can get the following formula:

$$
\begin{aligned}
\mathbb{E}_M\Big[\text{MaskingIG}_{t,d}(x, M) \;\Big|\; M_{t,d} = 1\Big] &= \mathbb{E}_M\Big[x_{t,d}\, M_{t,d} \int_0^1 \frac{\partial F_{\hat{y}}(z(\alpha; M))}{\partial x_{t,d}}\, d\alpha \;\Big|\; M_{t,d} = 1\Big] \\
&= x_{t,d} \int_0^1 \mathbb{E}_M\Big[\tfrac{\partial F_{\hat{y}}(z(\alpha; M))}{\partial x_{t,d}} \;\Big|\; M_{t,d} = 1\Big]\, d\alpha.
\end{aligned}
$$

$\square$

### B.2. Proof of Proposition 4.2

*Proof.* Since $x$ and $x'$ differ only at the single coordinate $(t, d)$, whenever $M_{t,d} = 1$, we have

$$
\tilde{x}(M) = (\mathbf{1} - M) \odot x + M \odot x' = x'.
$$

Hence, for any mask $M$ with $M_{t,d} = 1$, $\text{MaskingIG}_{t,d}(F, x, \tilde{x}(M))$ reduces to $\text{IG}_{t,d}(F, x, x')$. Taking the conditional expectation over all such $M$ yields

$$
\text{TIMING}_{t,d}(x) = \mathbb{E}\big[\text{MaskingIG}_{t,d}(F, x, \tilde{x}(M))\big|M_{t,d} = 1\big] = \text{IG}_{t,d}(x).
$$

By the sensitivity property of Integrated Gradients, if $F(x) \neq F(x')$, then $\text{IG}_{t,d}(x) \neq 0$. Consequently, $\text{TIMING}_{t,d}(x)$ must also be nonzero. $\square$

### B.3. Proof of Proposition 4.3

*Proof.* The TIMING formula depends only on the gradients of the model, similar to IG (Sundararajan et al., 2017). Therefore, it satisfies implementation invariance. $\square$

## C. Existing Evaluation Metrics for Time Series XAI

We introduce CPD and CPP as novel evaluation metrics designed to better capture model faithfulness. Consequently, we prioritize their use in our analysis while also employing existing XAI evaluation metrics for comparative purposes. For synthetic datasets, we assess feature importance using Area Under Precision (AUP) and Area Under Recall (AUR), alongside our proposed CPD metric. For real-world datasets, we adopt the evaluation criteria from (Enguehard, 2023; Liu et al., 2024b), detailed as follows:

- **Accuracy (Acc)**: Accuracy evaluates how often the model's original prediction is retained after removing salient features. A lower accuracy score indicates a more effective explanation.
- **Cross-Entropy (CE)**: Cross-Entropy quantifies the entropy difference between perturbed and original feature predictions, measuring information loss. A higher CE value is preferred.
- **Sufficiency (Suff)**: Sufficiency measures the average change in predicted class probabilities when only selected features are retained. A lower value is preferable, indicating minimal information loss.
- **Comprehensiveness (Comp)**: Comprehensiveness assesses how much the model's confidence in a target class decreases when important features are removed. A higher score suggests that the removed features were crucial for prediction, making it a stronger interpretability measure.

Traditional feature importance metrics, such as AUP and AUR, assume that the model correctly identifies ground truth importance in synthetic data. However, this assumption does not always hold in practice, limiting their reliability. While Acc, CE, Suff, and Comp are well-established metrics, they suffer from the drawback of *simultaneously removing multiple features*, potentially affecting interpretability. A natural extension of our cumulative metrics could involve integrating these conventional measures to develop a more robust evaluation framework.

## D. Description of Datasets

**Table 7:** We describe two synthetic datasets—Switch-Feature (Tonekaboni et al., 2020; Liu et al., 2024b) and State (Tonekaboni et al., 2020; Crabbé & Van Der Schaar, 2021)—and six real-world datasets—MIMIC-III (Johnson et al., 2016), PAM (Reiss & Stricker, 2012), Epilepsy (Andrzejak et al., 2001), Boiler (Shohet et al., 2019), Wafer (Dau et al., 2019), and Freezer (Dau et al., 2019)—which are all used in our experiments.

| Type | Name | Task | Dataset Size | Length | Dimension | Classes |
|------|------|------|--------------|--------|-----------|---------|
| Synthetic datasets | Switch-Feature | Binary classification | 1,000 | 100 | 3 | 2 |
| | State | Binary classification | 1,000 | 200 | 3 | 2 |
| | MIMIC-III | Mortality prediction | 22,988 | 48 | 32 | 2 |
| | PAM | Action recognition | 5,333 | 600 | 17 | 8 |
| Real-world datasets | Epilepsy | EEG classification | 11,500 | 178 | 1 | 2 |
| | Boiler | Mechanical fault detection | 90,115 | 36 | 20 | 2 |
| | Wafer | Wafer fault detection | 7,164 | 152 | 1 | 2 |
| | Freezer | Appliance classification | 3,000 | 301 | 1 | 2 |

Building on recent state-of-the-art studies (Liu et al., 2024b;a), we evaluate our method on 8 time series datasets spanning both synthetic and real-world domains, as summarized in Table 7.

### D.1. Synthetic Datasets

We consider 2 synthetic datasets: Switch-Feature (Tonekaboni et al., 2020; Liu et al., 2024b) and State (Tonekaboni et al., 2020; Crabbé & Van Der Schaar, 2021).

**Switch-Feature.** Following the protocol of Tonekaboni et al. (2020) and the implementation details in Liu et al. (2024b), we synthesize the dataset with a three-state hidden Markov model (HMM) whose initial distribution is $\pi = [\frac{1}{3}, \frac{1}{3}, \frac{1}{3}]$ and whose transition matrix is given as follows:

$$\begin{bmatrix} 0.95 & 0.02 & 0.03 \\ 0.02 & 0.95 & 0.03 \\ 0.03 & 0.02 & 0.95 \end{bmatrix}.$$

At each time step, the model occupies a state $s_t \in \{0, 1, 2\}$ and emits a three-dimensional feature vector $\mathbf{x}_t$ drawn from a Gaussian Process (GP) mixture. Each GP component employs an RBF kernel with $\gamma = 0.2$ and marginal variance 0.1; its mean vector is state-dependent:

$$\boldsymbol{\mu}_0 = [0.8, -0.5, -0.2], \quad \boldsymbol{\mu}_1 = [0, -1.0, 0], \quad \boldsymbol{\mu}_2 = [-0.2, -0.2, 0.8].$$

The binary label $y_t$ also depends on the state and is drawn from Bernoulli($p_t$) with

$$p_t = \begin{cases} 1/(1 + \exp(-\mathbf{x}_{t,0})) & \text{if } s_t = 0, \\ 1/(1 + \exp(-\mathbf{x}_{t,1})) & \text{else if } s_t = 1, \\ 1/(1 + \exp(-\mathbf{x}_{t,2})) & \text{else if } s_t = 2. \end{cases}$$

We generate 1,000 time series sequences of length 100 ($T = 100$) using the above procedure. The dataset is split into 800 training samples and 200 test samples for evaluating time series XAI methods. Each sample is annotated with the true saliency map $\{(t, s_t) \mid t = 1, \ldots, T\}$, as in Liu et al. (2024b).

**State.** First proposed by Tonekaboni et al. (2020) and later slightly modified by Crabbé & Van Der Schaar (2021), this dataset has since become a standard test-bed for time series XAI (Enguehard, 2023; Liu et al., 2024b). Each sequence is generated by a two-state hidden Markov model (HMM) with initial distribution $\pi = [0.5, 0.5]$ and transition matrix

$$\begin{bmatrix} 0.1 & 0.9 \\ 0.1 & 0.9 \end{bmatrix}.$$

Conditioned on the latent state $s_t \in \{0, 1\}$, an input feature vector $\mathbf{x}_t$ is sampled from $\mathcal{N}(\boldsymbol{\mu}_{s_t}, \Sigma_{s_t})$ with

$$\boldsymbol{\mu}_0 = [0.1, 1.6, 0.5], \quad \boldsymbol{\mu}_1 = [-0.1, -0.4, -1.5],$$

$$\Sigma_0 = \begin{bmatrix} 0.8 & 0 & 0 \\ 0 & 0.8 & 0.01 \\ 0 & 0.01 & 0.8 \end{bmatrix}, \quad \Sigma_1 = \begin{bmatrix} 0.8 & 0.01 & 0 \\ 0.01 & 0.8 & 0 \\ 0 & 0 & 0.8 \end{bmatrix}.$$

A binary label $y_t$ is then drawn from Bernoulli($p_t$) where

$$p_t = \begin{cases} 1/(1 + \exp(-\mathbf{x}_{t,1})) & \text{if } s_t = 0, \\ 1/(1 + \exp(-\mathbf{x}_{t,2})) & \text{else if } s_t = 1. \end{cases}$$

Using the above procedure, we synthesize 1,000 time series of length 200 ($T = 200$). For each sample, we define the true saliency map as $\{(t, 1 + s_t) \mid t = 1, \ldots, T\}$, following Crabbé & Van Der Schaar (2021). We train on the first 800 sequences and reserve the remaining 200 for evaluation.

### D.2. Real-world Datasets

We employ 6 real-world datasets: MIMIC-III (Johnson et al., 2016), PAM (Reiss & Stricker, 2012), Boiler (Shohet et al., 2019), Epilepsy (Andrzejak et al., 2001), Wafer (Dau et al., 2019), and Freezer (Dau et al., 2019).

**MIMIC-III.** Using the MIMIC-III database (Johnson et al., 2016), we assemble adult ICU admission data for in-hospital mortality prediction from the 48 hours of recorded data ($T = 48$). The data processing pipeline mirrors that of Tonekaboni et al. (2020), with two small tweaks: we restore a laboratory variable that was inadvertently dropped, and we apply a uniform constant fill to any missing entries. The final collection comprises 22,988 ICU admissions, each represented by 32 aligned clinical features ($D = 32$).

**PAM.** We adopt the Physical Activity Monitoring (PAM) dataset introduced by Reiss & Stricker (2012), which records 18 daily activities performed by 9 subjects wearing 3 inertial measurement units. Following the data processing protocol of Queen et al. (2024), we retain the 8 activities with at least 500 samples each. The resulting dataset comprises 5,333 samples, each consisting of 600 time steps ($T = 600$) across 17 sensor channels ($D = 17$).

**Table 8:** Comparison of model consistency across various XAI methods for MIMIC-III mortality prediction with zero substitution. We evaluate with Transformer and CNN models under 20% masking and cumulative masking strategies. Results (mean ± standard error) are averaged over five cross-validation repetitions, reporting multiple performance metrics, including accuracy (Acc) and cumulative prediction difference (CPD) at $K = 50$ and $K = 100$.

| Method | Transformer | | | CNN | | |
|---|---|---|---|---|---|---|
| | CPD ($K = 50$) ↑ | CPD ($K = 100$) ↑ | Acc ↓ | CPD ($K = 50$) ↑ | CPD ($K = 100$) ↑ | Acc ↓ |
| AFO | $0.063_{\pm 0.002}$ | $0.091_{\pm 0.004}$ | $0.975_{\pm 0.001}$ | $0.261_{\pm 0.021}$ | $0.424_{\pm 0.035}$ | $0.909_{\pm 0.008}$ |
| GradSHAP | $0.099_{\pm 0.004}$ | $0.126_{\pm 0.007}$ | $0.974_{\pm 0.001}$ | $0.855_{\pm 0.077}$ | $1.237_{\pm 0.111}$ | $0.903_{\pm 0.008}$ |
| Extrmask | $0.052_{\pm 0.002}$ | $0.081_{\pm 0.004}$ | $\underline{0.973_{\pm 0.002}}$ | $0.480_{\pm 0.037}$ | $0.633_{\pm 0.049}$ | $0.896_{\pm 0.009}$ |
| ContraLSP | $0.011_{\pm 0.001}$ | $0.024_{\pm 0.002}$ | $\mathbf{0.955_{\pm 0.005}}$ | $0.132_{\pm 0.007}$ | $0.313_{\pm 0.015}$ | $\mathbf{0.583_{\pm 0.015}}$ |
| TimeX | $0.049_{\pm 0.002}$ | $0.070_{\pm 0.003}$ | $0.974_{\pm 0.002}$ | $0.275_{\pm 0.022}$ | $0.396_{\pm 0.035}$ | $0.917_{\pm 0.009}$ |
| TimeX++ | $0.029_{\pm 0.001}$ | $0.040_{\pm 0.002}$ | $0.976_{\pm 0.001}$ | $0.329_{\pm 0.040}$ | $0.496_{\pm 0.068}$ | $0.914_{\pm 0.008}$ |
| IG | $0.103_{\pm 0.005}$ | $0.129_{\pm 0.007}$ | $0.974_{\pm 0.002}$ | $0.925_{\pm 0.087}$ | $1.349_{\pm 0.123}$ | $0.905_{\pm 0.007}$ |
| TIMING | $\mathbf{0.109_{\pm 0.005}}$ | $\mathbf{0.140_{\pm 0.008}}$ | $0.974_{\pm 0.002}$ | $\mathbf{1.173_{\pm 0.077}}$ | $\mathbf{1.826_{\pm 0.113}}$ | $\underline{0.874_{\pm 0.011}}$ |

**Boiler.** We adopt the Boiler dataset (Shohet et al., 2019), which contains multivariate sensor traces from simulated hot water heating boilers and is used to detect blowdown valve faults. Following the data processing pipeline of Queen et al. (2024), we obtain 90,115 samples, each represented by 36 time steps ($T = 36$) across 20 sensor channels ($D = 20$).

**Epilepsy.** The Epilepsy dataset (Andrzejak et al., 2001) consists of single-lead EEG recordings from 500 subjects. Following the preprocessing of Queen et al. (2024), we convert the original five classes into a binary label indicating whether a seizure occurs. Each subject is monitored for 23.6 seconds; the recording is segmented into 1 second windows sampled at 178 Hz, producing 11,500 samples. Each resulting sample comprises 178 time steps ($T = 178$) from a single channel ($D = 1$).

**Wafer and Freezer.** The Wafer and FreezerRegularTrain (Freezer) datasets are both drawn from the UCR archive (Dau et al., 2019) and are evaluated in Liu et al. (2024a) as standard benchmarks for time series classification. The Wafer dataset captures inline process control signals from semiconductor fabrication sensors. Each univariate time series contains 152 time steps ($T = 152$) recorded while processing a single wafer. The binary task is to detect abnormal wafers. The Freezer dataset comprises 3,000 univariate power-demand sequences, each 301 time steps long ($T = 301$), collected from two household freezers located in a kitchen and a garage. The classification task is to determine which freezer generated each sequence.

## E. Result on Different Black-Box Classifiers

Our main experiments focus on a single-layer GRU with 200 hidden units as the primary model architecture. To further validate the generalizability of our approach, we extended our black box models to include Convolutional Neural Network (CNN) and Transformer (Vaswani et al., 2017), as suggested in TimeX++ (Liu et al., 2024a). As illustrated in Table 8, TIMING can generalize across the type of black box model.

## F. Qualitative Examples

Due to space limitations in the main text, all of the qualitative figures and analysis for Figure 10 are included in this appendix. In Figure 10, the visualization of feature attributions across methods highlights key differences in how signed and unsigned methods identify salient features. Signed methods like TIMING, GradSHAP (Lundberg & Lee, 2017), DeepLIFT (Shrikumar et al., 2017), and IG (Sundararajan et al., 2017) consistently assign importance to feature index 9 (lactate) in specific regions, aligning with clinical knowledge that elevated lactate levels contribute to lactic acidosis, a condition strongly linked to mortality. In contrast, unsigned methods fail to clearly identify lactate as a salient feature, suggesting limitations in their ability to explain model behavior and align with clinical understanding.

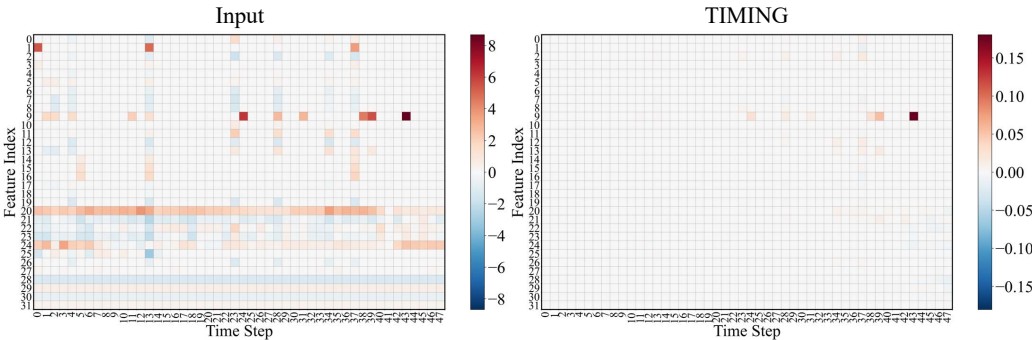

**Figure 6:** Qualitative analysis of input features and attributions extracted from TIMING on the MIMIC-III mortality benchmark (Johnson et al., 2016) for a true positive case where (label = 1, model output = 0.625).

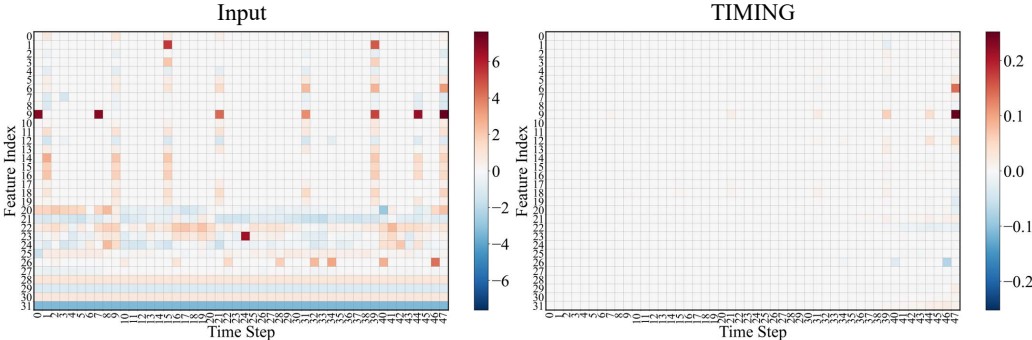

**Figure 7:** Qualitative analysis of input features and attributions extracted from TIMING on the MIMIC-III mortality benchmark (Johnson et al., 2016) for a true positive case where (label = 1, model output = 0.898).

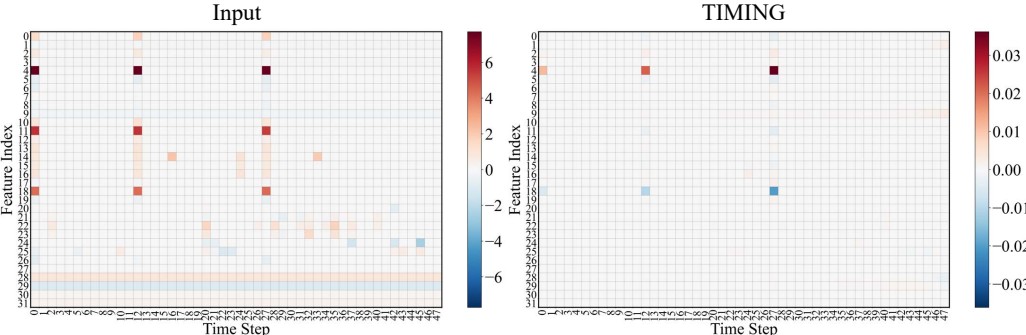

**Figure 8:** Qualitative analysis of input features and attributions extracted from TIMING on the MIMIC-III mortality benchmark (Johnson et al., 2016) for a true negative case where (label = 0, model output = 0.020).

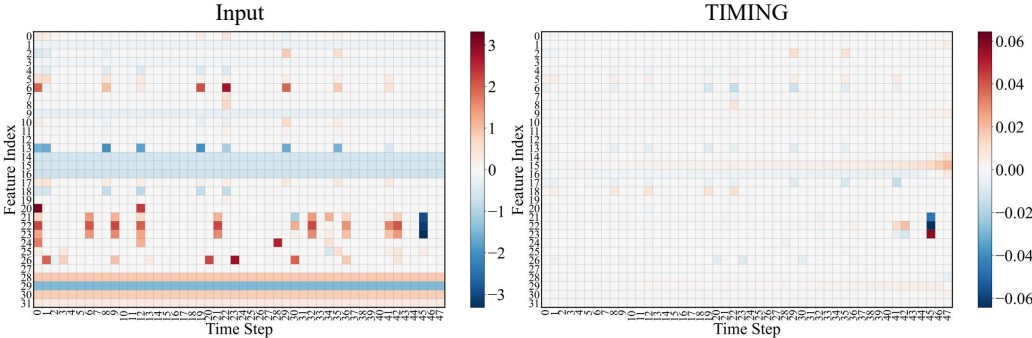

**Figure 9:** Qualitative analysis of input features and attributions extracted from TIMING on the MIMIC-III mortality benchmark (Johnson et al., 2016) for a true negative case where (label = 0, model output = 0.081).

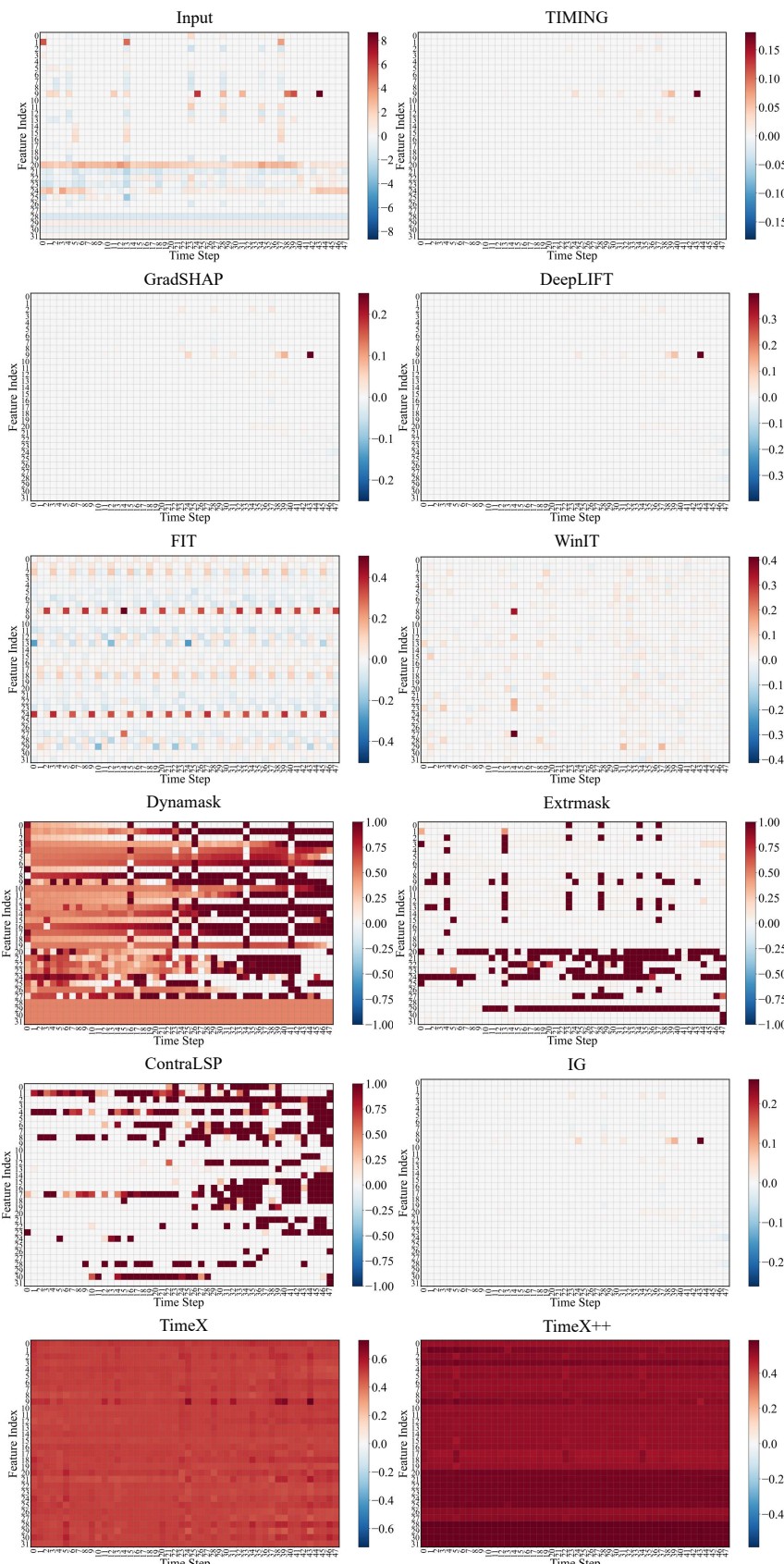

**Figure 10:** Qualitative analysis of input features and attributions extracted from TIMING and baselines on the MIMIC-III mortality benchmark (Johnson et al., 2016) for a true positive case where (label = 1, model output = 0.625).

