# OpenReview forum: "TIMING: Temporality-Aware Integrated Gradients for Time Series Explanation"
_ICML.cc/2025/Conference — ICML 2025 spotlightposter_

### Official Review · Reviewer_ynvH · 2025-02-22

**Overall Recommendation:** 3

**Summary:**

The paper first reveals that Integrated Gradients (IG) effectively captures important points but has been underestimated in previous research due to traditional evaluation metrics' inability to consider feature importance's directional information. Therefore, the authors propose novel evaluation metrics, CPD and CPP, which comprehensively assess attribution methods and re-evaluate baselines. Then, based on the effectiveness of directional attribution methods, they introduce Time Series Integrated Gradients (TIMING) for time series data. TIMING overcomes the limitations of traditional IG in handling complex temporal dependencies and out-of-distribution issues by integrating temporality-aware dynamic baselines into the attribution calculation and demonstrating through extensive experiments that TIMING outperforms baselines while maintaining IG's efficiency and theoretical properties.

**Claims And Evidence:**

Yes, the evidence is clear.

**Essential References Not Discussed:**

No. All the prior related references that seem relevant to the proposed evaluation metrics (CPD and CPP), the improvement of IG for time series data (TIMING), and the issues with existing evaluation methods and limitations of applying IG to time series data have been appropriately covered and cited within the paper.

**Experimental Designs Or Analyses:**

The experiment is complete, and most of the experiments were conducted on MIMIC-III.
My question is why gradient-based methods (IG, GradSHAP ...) have lower CPD and CPP. Is it because their highlighted important features are too sparse, like Figure 10? If only a small number of features are considered, is there a potential risk?

**Methods And Evaluation Criteria:**

The proposed method and evaluation criteria are well-aligned with the challenges of time series XAI. CPD and CPP provide nuanced evaluation metrics, while TIMING offers a theoretically grounded, temporally adaptive solution based on IG.
However, there are several points that need to be explained:

* It is unclear how the random retention strategy can solve the problem of OOD (line r213). I have reviewed Table 5, but it does not effectively indicate that the OOD issue has been alleviated.
* What if the mask $M\in [0,1]$ is a soft mask for sampling.
* It seems that all masks $M_{t,d}$ are independent and identically distributed for each feature. Should we consider temporal continuity in explanation? such as a simple consideration, $|M_{t,d} - M_{t+1,d}|<\theta$.
* Is there any relationship between CPD and AUP, and how to improve both metrics simultaneously?
* Why does a big $n$ in Table 7 actually have a worse effect, and how should we select $n$ for a specific real dataset?

**Other Comments Or Suggestions:**

Does the definition of CPD and CPD need to be distinguished in the formula (lines r123 and r141)?

**Other Strengths And Weaknesses:**

The paper makes valuable contributions by re-evaluating and enhancing gradient-based methods for time series XAI, challenging prior norms, and introducing principled metrics. While some aspects of TIMING’s novelty may be similar to existing IG with ensemble learning, the work meaningfully advances the field’s understanding of directional attribution in temporal data.

**Questions For Authors:**

Please see the methods and evaluation criteria.

**Relation To Broader Scientific Literature:**

The paper’s contributions build upon and address limitations in the time series explainable AI (XAI) literature. Prior work (Tonekaboni et al., 2020; Crabbe & Van Der Schaar, 2021; Enguehard, 2023) focused on unsigned attributions, emphasizing feature importance magnitudes but ignoring directional effects (positive/negative contributions). This contrasts with signed attribution methods like Integrated Gradients (IG) (Sundararajan et al., 2017), widely used in non-temporal domains, which inherently encode directionality. However, prior evaluations of time series XAI methods (e.g., simultaneous masking of high-attribution points) failed to account for directional cancellation effects, leading to underestimation of IG’s efficacy. The proposed CPD/CPP metrics address this gap by tracking incremental attribution impacts, resolving the cancellation issue and aligning with evaluation principles.

**Theoretical Claims:**

The method is based on Integrated Gradients, so there is no specific theory to check it.

---

> ### Author Rebuttal · Authors · 2025-04-01
>
> We greatly appreciate your valuable feedback and have responded to each of your questions below.
>
> ---
>
> [Q1] Random retention strategy to mitigate OOD.
>
> [A1] Thank you for this comment. Integrated Gradients (IG) often interpolates through out-of-distribution (OOD) regions. Our proposed random retention strategy (RandIG) addresses two key issues concurrently: it reduces OOD occurrences by partially retaining original inputs and captures cases involving disrupted temporal dependencies, which are especially critical in time series contexts. Table 5 illustrates RandIG’s improved performance over IG in addressing these concerns. Furthermore, our TIMING method equipped with segment-based masking strategy preserves segment-level information, thus more effectively mitigating the OOD issue compared to RandIG, as shown in Table 5.
>
> ---
>
> [Q2] Soft Mask ($M \in [0,1]$) & Temporal Continuity.
>
> [A2] Our segment-based sampling approach generates binary masks with inherent temporal continuity. Unlike independent sampling (RandIG), our method preserves local temporal structure, which enhances overall performance by maintaining contextual relationships between contiguous segments.
>
> While soft masks ($M \in [0, 1]$) show promise, we currently face challenges in handling gradients for fractional mask values. Our current method relies on gradients from zero-masked positions, and integrating soft masks would require sophisticated theoretical and computational approaches, marking an important avenue for future research.
>
> ---
>
> [Q3] Relationship between CPD and AUP.
>
> [A3] CPD measures feature influence on model predictions, while AUP evaluates alignment with the data generation process. In XAI methods, explaining model behavior is the primary focus.
>
> Higher CPD doesn't always correlate with higher AUP. When models focus on different features, more accurate attribution might actually decrease AUP. This complexity reflects real-world challenges in model interpretation, where perfect alignment with data generation processes is rare.
>
> TIMING demonstrates model faithfulness by prioritizing accurate model explanations over detecting specific data features, highlighting the nuanced nature of explanation methods.
>
> ---
>
> [Q4] Worse effect of a big n and selection criteria.
>
> [A4] Thank you for your question. In Table 7, while $(n, s_{min}, s_{max}) = (100, 10, 48)$ results in a lower score, this is simply a natural outcome of using a large $n$: when n is large, many points remain each step in our modified baseline $(1 - M) \odot x $, so some points receive few (or no) gradient calculations over the path. Consequently, attribution quality can appear lower.
>
> Regarding how to choose $n$ for real-world datasets, our hyperparameter tuning suggests the following when $n_{samples}=50$:
>
> - Multivariate: $(n, s_{min},s_{max})=(2D, 1, T)$ often performs well.
> - Univariate: ensure that $2n \times s_{max} \approx T$.
>
> Nonetheless, as discussed in our paper, TIMING is robust to different hyperparameter choices. The main principle is to avoid retaining too many points in the masking, which can undermine its effectiveness.
>
> ---
>
> [Q5] Reason for gradient-based methods’ better performance – due to sparsity?
>
> [A5] Thank you for this insightful question. We believe these high CPD and low CPP scores are not simply due to sparsity. If that were the case, their advantage would likely drop when shifting from K=50 to K=100, since adding more features makes it harder to rely on only a few key points. Instead, in our MIMIC-III experiments, they actually maintain or improve performance in the K=100 setting across various architectures. Additionally, these methods stay robust under 20% masking across multiple datasets, suggesting they truly identify features that drive model predictions, rather than relying on sparsity alone.
>
> ---
>
> [Q6] Performance of black-box-classifiers.
>
> [A6] Thank you for your question. We will include the requested results in the Appendix of our paper. We also note that the significantly improved performance on certain datasets (e.g., Boiler), compared to results reported in TimeX++, appears to be largely due to the normalization step in our data preprocessing.
>
> - [Table for the single-layer GRU with 200 hidden units.](https://postimg.cc/kBD66nYm)
> - [Table for different black-box classifier.](https://postimg.cc/0MzmgDMb)
>
> ---
>
> [Q7] Formula should distinguish between CPD and CPP definitions.
>
> [A7] We agree that distinguishing these two definitions would improve clarity. We propose the following notations:
>
> $$
> \begin{equation*} \text{CPD}(x) = \sum_{k=0}^{K-1} \left\| F(x^{\uparrow}_k) - F(x^{\uparrow}\_{k+1}) \right\|_1, \end{equation*}
> $$
>
> $$
> \begin{equation*} \text{CPP}(x) = \sum_{k=0}^{K-1} \left\| F(x^{\downarrow}_k) - F(x^{\downarrow}\_{k+1}) \right\|_1, \end{equation*}
> $$
>
> where $x_k^{\uparrow}$ and $x_k^{\downarrow}$ refer to the input after the removal of the top-k points with the highest and lowest absolute attributions, respectively.

---

### Official Review · Reviewer_sX8p · 2025-03-01

**Overall Recommendation:** 3

**Summary:**

The paper addresses the explainable AI (XAI) issue in time series. It proposes CPD and CPP evaluation metrics, discovers that traditional Integrated Gradients (IG) performs well, and then presents the TIMING method. Experiments show that TIMING outperforms baseline methods in multiple aspects.

**Claims And Evidence:**

Most claims are supported by evidence, but the claim that TIMING's segment-based masking is the best lacks sufficient proof.

**Essential References Not Discussed:**

No essential references are missing, and the literature review in the paper is comprehensive.

**Experimental Designs Or Analyses:**

The experimental designs are sound, comparing with 12 baseline methods and conducting ablation studies.

**Methods And Evaluation Criteria:**

The methods and evaluation criteria are reasonable. CPD, CPP, and TIMING's segment - based masking suit the needs of time series XAI, and the evaluation with multiple datasets and metrics is comprehensive.

**Other Comments Or Suggestions:**

The paper should further explore the practical impact of TIMING's incompleteness, add real-world case studies.

**Other Strengths And Weaknesses:**

- Strengths: The evaluation metrics and the TIMING method are novel, the experiments are comprehensive, and the theoretical basis is solid.
- Weaknesses: The generalization of TIMING needs to be enhanced, and the focus on directional attributions may be unnecessary in some tasks.

**Questions For Authors:**

- Q1: Have you considered comparing with more complex masking strategies? What would the results be?
- Q2: How do you suggest users interpret the differences between TIMING attributions and model outputs?

**Relation To Broader Scientific Literature:**

The proposed metrics and methods are based on existing research and improve time series XAI.

**Theoretical Claims:**

The proofs of other theoretical properties assume certain conditions that may not hold in all practical situations, weakening the theoretical foundation of TIMING.

---

> ### Author Rebuttal · Authors · 2025-04-01
>
> We sincerely thank you for your helpful feedback and have addressed each of your comments below.
>
> ---
>
> [Q1] Insufficient evidence for TIMING's segment-based masking superiority & comparison with complex masking.
>
> [A1] Our results across real-world (Table 2, 3) and synthetic (Table 4) datasets show that TIMING consistently outperforms IG, demonstrating the practical benefits of segment-based masking within the integration path. The ablation study in Table 5 further confirms that our "segment-based" masking approach, which considers temporal dependencies by simultaneously masking consecutive points, surpasses RandIG's random masking of individual points.
>
> While our segment-based random masking performs well, it could be enhanced through "data-dependent" strategies:
>
> - Replacing masked portions with counterfactuals to calculate attributions as points contribute in the "opposite direction."
> - Mining temporal motifs or shapelets and adjusting masking probabilities to preserve these patterns, analyzing how breaking typical patterns affects point contributions.
>
> These approaches offer promising directions to refine our segment-based masking and better capture temporal dependencies.
>
> ---
>
> [Q2] Proofs rely on assumptions not always valid in practice, weakening TIMING’s theoretical foundation.
>
> [A2] Thank you for the constructive feedback. TIMING relies on three assumptions:
>
> - (S1) Partial derivatives are bounded along all of the paths from $\tilde{x}(M)$ to $x$.
> - (S2) $M$ follows a probability distribution.
> - (S3) $Pr(M_{t,d}=1)>0$.
>
> We agree that assumption S1 might raise concerns in practical scenarios. However, it's commonly adopted in ML theory [1][2] and typically holds for modern neural networks. Moreover, these assumptions primarily serve to justify that multiple Integrated Gradients (IG) computations can efficiently share a single IG path. Therefore, any potential gradient divergence would impact IG in general, rather than specifically undermining TIMING’s theoretical foundation. Assumptions S2 and S3 are practically reasonable. If any assumptions remain problematic or unclear, please let us know.
>
> [1] Diederik P. Kingma and Jimmy Ba. "Adam: A Method for Stochastic Optimization." ICLR 2015.
>
> [2] Bernstein et al. "signSGD: Compressed Optimisation for Non-Convex Problems." ICML 2018.
>
> ---
>
> [Q3] Weakness: Limited generalization of TIMING; directional attribution not always necessary.
>
> [A3] TIMING inherits gradient-based feature attribution estimation from Integrated Gradients (IG). As a general XAI method, TIMING can be broadly applied to time series. It is important to note that TIMING is not specialized solely in direction estimation, but naturally provides both magnitude and direction.
>
> Our experiments support this. For fair comparison with undirectional state-of-the-art time series XAI methods, we "absolutized" our attributions to compare magnitude estimation. Even so, TIMING achieved superior SOTA performance. Thus, TIMING outperforms methods focusing only on magnitude and offers the additional benefit of directional information.
>
> ---
>
> [Q4] Explore practical impact of TIMING's incompleteness; add real-world examples.
>
> [A4] Thank you for your valuable feedback about the practical impact of TIMING's theoretical incompleteness and for suggesting additional real-world case studies.
>
> While completeness is a desirable theoretical property, our research demonstrates that TIMING's approach prioritizes accurately identifying the most influential features in time series data, which is typically more critical in practical applications. For instance, in healthcare scenarios, correctly identifying influential features is crucial for doctor and patient to trust the model, whereas completeness may be of lesser practical significance.
>
> ---
>
> [Q5] Interpretation of differences between TIMING attributions and model outputs.
>
> [A5] Thank you for this important question. In standard Integrated Gradients (IG), due to completeness, attributions directly represent individual contributions summing exactly to $f(x) - f(x')$. However, these standard IG attributions are less reliable for time-series data.
>
> In TIMING, without the condition $M_{t,d}=1$, the attribution reflects the expected model-output difference over various masked samples:
>
> $
> \sum_{t,d}\mathbb{E}\_{M \sim G(n, s_{min}, s_{max})}[\text{MaskingIG}\_{t,d}(x,M)] = \mathbb{E}\_{M \sim G(n, s_{min}, s_{max})}[\sum_{t,d}\text{MaskingIG}\_{t,d}(x,M)] = \mathbb{E}\_{M \sim G(n, s_{min}, s_{max})}[f(x)-f((\mathbf{1}-M) \odot x)]
> $
>
> Thus, users can interpret TIMING’s attributions as the expected influence of each element over all baselines retaining specific segments. However, without conditioning on $M_{t,d}=1$, attribution values may become biased due to variations in masking frequency. By imposing $M_{t,d}=1$, TIMING ensures consistent evaluation of each feature’s contribution, resulting in more accurate and reliable interpretations for time-series data.

---

### Official Review · Reviewer_Mpsu · 2025-03-13

**Overall Recommendation:** 4

**Summary:**

This paper proposes an improved version of the integrated gradient for time series tasks. The paper also challenges previous metrics for evaluating time series explainability and accordingly proposes two improved metrics for better evaluation. Overall, I think it is a good paper.

## update after rebuttal

Thanks for the response, which addressed my concern and helped me better understand the paper. So, I decided to raise my score to 4.

**Claims And Evidence:**

The authors claim that simultaneously masking out will raise issues in time series data, which makes sense to me. Their motivation is good, but I am still trying to understand Fig 1.

The proposed method also supports their motivation, but if I understand correctly, authors only use random segmentation masking to characterize the temporal dependencies?

**Essential References Not Discussed:**

Can you also include the recent time series multiple-instance learning framework in the discussion? Since I noticed they mentioned they have somehow explained abilities?

- Inherently interpretable time series classification via multiple instance learning, ICLR'24
- TimeMIL: Advancing multivariate time series classification via a time-aware multiple instance learning, ICML'24

**Experimental Designs Or Analyses:**

- The experiments followed some standard flow consistency with the previous works, which I believe is sound.
- From my observation, IG is almost better than other baselines. Can authors comment on this?

**Methods And Evaluation Criteria:**

Yes.

**Other Comments Or Suggestions:**

One concern is this method may rely on the segment window size and number of windows. I saw authors conduct such ablation, but is there possibly a better way to generate the window?

**Other Strengths And Weaknesses:**

See above.

**Questions For Authors:**

- What is the y-axis in Fig.1?
- Can you elaborate more about the ground truth? How is it formatted? Can it be visualized?

**Relation To Broader Scientific Literature:**

This paper made a bridge between IG and the time series community.

**Theoretical Claims:**

Yes. The theoretical analysis in the paper mostly reaffirms known properties of IG rather than introducing new insights.

---

> ### Author Rebuttal · Authors · 2025-04-01
>
> We appreciate your thoughtful insights and have addressed each of your comments individually below.
>
> ---
>
> [Q1] Clarification on Figure 1. (particularly its y-axis)
>
> [A1] Figure 1 has two components. The upper portion shows ground truth signed attributions on the y-axis. Two XAI methods estimate attributions, with features arranged by absolute importance. Method (a) has perfect feature selection by absolute value but alternating signs, while method (b) has consistent signs but inaccurate absolute values.
>
> The lower portion shows evaluation results after removing top-K features. Panel (c), the conventional prediction difference metric, suffers from cancellation effects, making method (b) appear superior due to sign alignment. Panel (d), our cumulative prediction difference, compares consecutive predictions to mitigate cancellation and enable accurate comparison.
>
> We appreciate your feedback and will clarify this in the final draft. Let us know if you have further questions.
>
> ---
>
> [Q2] Using only random segmentation to characterize the temporal dependencies?
>
> Thank you for your observation. Your understanding is correct. As stated in our paper, we began with a random retaining strategy to simulate scenarios where temporal dependencies are disrupted while keeping intermediate path values close to x to address out-of-distribution issues.
>
> Building on that, we introduced segment-based masking to better suit time series data. Retaining several segments helps preserve segment-level information, allowing the model to handle both preserved and disrupted temporal relationships. Although this approach is simple, we find it powerful for time series.
>
> Nonetheless, customizing the masking strategy for specific datasets or developing more complex methods remains a promising direction for future work.
>
> ---
>
> [Q3] IG outperforms other baselines, any comments?
>
> [A3] Your observation aligns with our central hypothesis: gradient-based methods like IG, DeepLIFT, and GradSHAP can identify both positively and negatively important points (similar to method (a) in Fig. 1), but conventional evaluation metrics underestimate their effectiveness due to cancellation effects when simultaneously removing top-K points.
>
> Our CPP and CPD metrics address this limitation by preventing cancellation during evaluation. Results confirm that gradient-based baselines outperform state-of-the-art time series XAI methods like TimeX++ and ContraLSP when evaluated with metrics that avoid cancellation.
>
> Also, TIMING improves upon state-of-the-art results by incorporating a random masking strategy during the IG path. This novel approach leads to practical performance enhancements, as demonstrated by our experimental results.
>
> ---
>
> [Q4] Comparision to recent time series multiple-instance learning framework.
>
> [A4] Thank you for pointing out multiple instance learning approaches for time series interpretability [1][2]. While they are indeed relevant, they rely on specific ante-hoc explainable architectures rather than providing model-agnostic post-hoc explanations. As a result, we did not include them as our baselines. However, we recognize their importance for time series interpretability and will incorporate them in the related works section.
>
> [1] Early et al., "Inherently Interpretable Time Series Classification via Multiple Instance Learning." ICLR 2024.
>
> [2] Chen et al. "TimeMIL: Advancing Multivariate Time Series Classification via a Time-aware Multiple Instance Learning." ICML 2024.
>
> ---
>
> [Q5] Concern about dependency on window hyperparameters & alternatives?
>
> [A5] Indeed, our method currently depends on segment window size and the number of windows. However, as demonstrated by our ablation study, TIMING is robust and maintains stable performance across a wide range of segment configurations, significantly mitigating sensitivity concerns. Additionally, as discussed in Reviewer ynvH-[A4], we only need to carefully consider cases involving the retention of too many points during masking.
>
> Nonetheless, we agree that adaptive window generation could offer further advantages. As provided in Reviewer sX8p-[A1], exploring such adaptive or data-driven window methods is a promising direction for future work.
>
> ---
>
> [Q6] Clarify the format and visualization of ground truth.
>
> [A6] The ground truth in our synthetic datasets consists of binary labels explicitly marking input points responsible for output generation. These labels are clearly defined during data construction, enabling direct visualization of ground truth saliency maps. However, as mentioned in our paper, it may differ from what the trained model actually considers important, even if the model achieves high accuracy.
>
> Since our approach exactly follows ContraLSP, please refer to their visualization and construction process detailed in Figures 12–13 and Appendix D.2 of that work. We will include additional detailed explanations and visualizations of ground truth construction in the appendix.

---

### Official Review · Reviewer_ed7s · 2025-03-14

**Overall Recommendation:** 4

**Summary:**

The authors introduce Temporality-Aware Integrated Gradients which addresses the reliability issues of naive IG in the time series setting by applying a random retraining to partially retain certain data points with a segment-based mask. The theoretical properties of this approach are explored and comprehensive evaluation is provided. Perhaps the more helpful contribution is the introduction of the cumulative prediction difference and cumulative prediction preservation metrics which provide more fair comparison between attribution methods for time series. These evaluations show that in real-world tasks like MIMIC-III the original interpretability methods like IG and GradSHAP can still perform quite well. This work highlights once again how difficult it is to accurately evaluate time series interpretability methods and continues to advance the field.

**Claims And Evidence:**

The authors put forward a number of well-substantiated claims

C1. Current interpretability evaluation metrics are limited because of their focus on magnitude rather than direction of attribution or focusing on performance drop which shifts the explanatory focus to the data. This claim of limitation is a hard one to substantiate - and perhaps is best done in conjunction with evaluation of human-in-the-loop interpretability. However the measurement of sign-aligning bias is a reasonable way to show the misdirected attribution of current evaluations.

C2. TIMING outperforms baseline methods on proposed CPD/CPP metrics across multiple datasets. This is well evaluated and the results are presented clearly in the paper

C3. TIMING is computationally efficient and competitive with standard interpretability methods like GradSHAP/IG for its performance. This is validated with elapsed time on MIMIC-III though no theoretical analysis of time/memory complexity is completed.

**Essential References Not Discussed:**

All the primary and essential works seem to be covered. Some adjacent work on applying multiple instance learning to time series interpretability and the theoretical implications of that work for evaluating interpretability methods could be added. Early et. al. (2023) could be a good place to start for this. Additionally, engaging with counterfactual and human-centered interetability methods could help expand the applications of this work.

**Experimental Designs Or Analyses:**

Comparative evaluation showed TIMING performance against up to 12 other methods on the synthetic and real-world tasks. The authors compared CPD, CPP, accuracy, cross-entropy, sufficiency, and complexity as well as CPD under different substitution strategies. For synthetic datasets AUP/AUR was also considered.

Isolated effects of segment-based masking versus random point masking were evaluated with ablation studies. In additional to the primary evaluations on GRU some CNN and Transformer experiments were done to show generalization across model types. Hyperparameter sensitivity was also evaluated.

Computational efficiency was analyzed in terms of raw time, although this analysis could be expanded on to evaluate the theoretical compute complexity of the algorithm.

Qualitative review of explanations generated by TIMING was shown in the appendices.

**Methods And Evaluation Criteria:**

1/ CPD and CPP metrics: These metrics address a real limitation in existing evaluation approaches by sequentially removing features rather than simultaneously, preventing the cancellation of positive and negative attributions. This approach aligns better with how models actually use features.
2/ TIMING algorithm: The method enhances IG by incorporating temporal dependencies through segment-based masking, which is appropriate for time series data where relationships between points matter. The randomization strategy mitigates out-of-distribution issues in the integration path.
3/ Evaluation approach: The authors evaluate on both synthetic (Switch-Feature, State) and real-world datasets (MIMIC-III, PAM, Boiler, etc) across multiple domains, using both their proposed metrics and conventional metrics. The comparison against 12 baseline methods is comprehensive. It could be helpful to understand more about the selection criteria for the task as some tasks from various sources have been left out (e.g. delayed spike from Leung et. al., seqcomb/freqshapes from Liu et al. (2024b) and incorrectly attribute the state dataset to Liu et al. when it was actually developed by Tonekaboni et al.
4/ Ablation studies: The ablation study in Table 5 effectively shows the value of segment-based masking over point-wise masking.

**Other Comments Or Suggestions:**

N/A

**Other Strengths And Weaknesses:**

S1. The theoretical analysis connecting TIMING to IG is rigorous and backs up the empirical evidence provided.
S2. The identification of sign cancellation in evaluation metrics is an important insight.
S3. Experiments are completed comprehensively across diverse datasets.

W1. Absence of comparison to other explainability paradigms beyond attribution methods.

**Questions For Authors:**

Q1. Did the authors consider whether other gradient-based methods suffer from similar problems to IG and if these temporality modifications could apply more broadly to that class of interpretability methods?

Q2. How could this method be extended to the time series regression setting?

**Relation To Broader Scientific Literature:**

The paper positions its work within the broader research of model interpretability for time series. It compares modality-agnostic methods (FO, AFO, GradSHAP, DeepLIFT, LIME) and time-series approaches (FIT, WinIT, Dynamask, Extrmask, ContraLSP, TimeX++).

**Theoretical Claims:**

TIMING maintains theoretical properties of IG around sensitivity and implementation invariance, though not completeness. These claims have been proven in the appendices. I reviewed the proofs and did not see any issues. Proposition 4.4 can be supported through the addition of an appropriate counterexample for additional thoroughness and seems an adequate tradeoff for gaining temporal awareness and the added performance benefit.

---

> ### Author Rebuttal · Authors · 2025-04-01
>
> We sincerely thank you for your insightful feedback. Below, we address each comment individually.
>
> ---
>
> [Q1] C3. Theoretical analysis of time/memory complexity.
>
> [A1] As shown in Fig. 4, TIMING demonstrates high efficiency compared to baselines like LIME, FO, AFO, and modern time series XAI methods, which rely on expensive forward passes or mask optimization (Compare IG and TIMING in Figure 4).
>
> The algorithm's time complexity is $O(b + m)$, with $b$ representing forward/backward pass cost and $m$ for mask sampling. Parallelizing integration steps significantly enhances efficiency, with mask sampling cost remaining negligible compared to gradient computation.
>
> Memory complexity of $O(n_{samples} \times (T \times D + B))$ matches Integrated Gradients, ensuring TIMING maintains high efficiency while providing superior attribution quality for time series explanation tasks.
>
> ---
>
> [Q2] Selection criteria for the task.
>
> [A2] We based our task selection on established benchmark datasets from ContraLSP (the previous SOTA method). To expand to diverse real-world datasets, we also included datasets from Table 4 of TimeX++ (another SOTA time series XAI work).
>
> We prioritize practical utility over ground truth identification. Conventional metrics like AUP and AUR assume perfect feature learning, which is unrealistic, as models can predict correctly by focusing on different features.
>
> Our CPD and CPP metrics provide a more nuanced evaluation of attributions. Even on synthetic datasets, our method demonstrates superior performance, justifying our use of the two classification datasets from ContraLSP for synthetic data evaluation.
>
> ---
>
> [Q3] Attribution correction for the state dataset. (Liu et al. to Tonekaboni et al.)
>
> [A3] We initially cited ContraLSP (Liu et al.) for the state dataset since they utilized it in their experiments. However, as the reviewer suggests, the dataset was actually developed by Tonekaboni et al; it was further modified in Dynamask (Crabbe & Van Der Schaar), which was subsequently adopted by Extrmask, ContraLSP, and our paper. We will revise the citations to FIT (Tonekaboni et al.) and Dynamask (Crabbe & Van Der Schaar) in our final draft.
>
> ---
>
> [Q4] Strengthening Proposition 4.4 with a counterexample.
>
> [A4] Thank you for this valuable suggestion. As noted in Proposition 4.4, developing IG variants for time series indeed leads to sacrificing completeness. However, we argue this trade-off is justified in scenarios dependent on temporal relationships.
>
> For clarity, consider a scenario involving $x=(x_1,x_2,x_3)$ where $x_1<x_2<x_3$ and function $f(x)=(x_3 - x_2)I(x_3>x_2)+(x_2 - x_1)I(x_2>x_1)$. Standard IG returns zero attributions for $x_2$ due to its interpolation path structure. Contrastingly, TIMING explores multiple partial interpolation paths, identifying meaningful gradients for $x_2$ and thus reflecting temporal dynamics more accurately. We will integrate a refined counterexample into our final version accordingly.
>
> ---
>
> [Q5] Comparision to other explainability paradigms.
>
> [A5] Related to multiple instance learning, please refer to reviewer Mpsu-[A4].
>
> Regarding your suggestions about counterfactual and human-centered methods, we agree these are valuable directions. That said, they extend beyond our current focus on time series–specific, model-agnostic XAI. We appreciate your feedback and may explore these broader approaches in future studies.
>
> ---
>
> [Q6] Extension of temporality modifications to other methods.
>
> [A6] Thank you for this insightful question. Our main focus was addressing issues of Integrated Gradients (IG) in time-series contexts, leading to the development of TIMING. As suggested, we additionally tested our masking strategy on other gradient-based methods by adjusting baselines and computing approximations of expectations.
>
> DeepLIFT’s CPD scores remain unchanged (only -0.001 at K=100), whereas GradSHAP exhibited slight improvements (+0.016 at K=50; +0.020 at K=100), likely due to its similarity to IG.
>
> We suspect that our segment-retaining approach is especially well-suited for IG, and other interpretability methods might require tailored solutions. Nonetheless, we believe these temporality modifications could be broadly beneficial and consider their extension an exciting direction for future work.
>
> ---
>
> [Q7] Extension to time series regression.
>
> [A7] TIMING naturally extends to regression settings by generating attributions for each time point in parallel. When the output is a vector, the method creates a comprehensive attribution map that measures each point's contribution to individual output elements.
>
> [Experiments on the MLO-Cn2 dataset](https://postimg.cc/McYK3435) [1] verify TIMING's superior CPD performance compared to existing baselines, confirming that our approach extends correctly to regression.
>
> [1] Jellen et al. "Effective Benchmarks for Optical Turbulence Modeling." arXiv preprint, 2024.

---

### Decision · Program_Chairs · 2025-05-01

**Decision:**

Accept (spotlight poster)

**Comment:**

Submission 15165 works on the problem of time-series attribution. They highlight that existing metrics do not fully capture the pos/neg impact on model predictions. To address this, they introduce two new metrics, called Cumulative Prediction Difference (CPD) and Cumulative Prediction Preservation (CPP), which better evaluate directional attribution. To further improve IG for time series, they propose **timing**, a temporal-aware extension to Integrated Gradients (IG)that respects time structure while preserving IG's theoretical guarantees. Results on synthetic and real-world data show their approach achieves state-of-the-art performance. The authors should add the related works and clarifications the reviewers raised in their camera ready.